# LIT-LVM: Structured Regularization for Interaction Terms in Linear Predictors using Latent Variable Models

**Mohammadreza Nemati**                                              *mohammadreza.nemati@case.edu*
*Department of Computer and Data Sciences*
*Case Western Reserve University*

**Zhipeng Huang**[*]                                                  *zhipeng.huang@suffolk.edu*
*Department of Mathematics and Computer Science*
*Suffolk University*

**Kevin S. Xu**                                                      *ksx2@case.edu*
*Department of Computer and Data Sciences*
*Case Western Reserve University*

**Reviewed on OpenReview:** *https://openreview.net/forum?id=3uW5nxESu1*

## Abstract

Some of the simplest, yet most frequently used predictors in statistics and machine learning use weighted linear combinations of features. Such linear predictors can model non-linear relationships between features by adding interaction terms corresponding to the products of all pairs of features. We consider the problem of accurately estimating coefficients for interaction terms in linear predictors. We hypothesize that the coefficients for different interaction terms have an *approximate low-dimensional structure* and represent each feature by a latent vector in a low-dimensional space. This low-dimensional representation can be viewed as a *structured regularization* approach that further mitigates overfitting in high-dimensional settings beyond standard regularizers such as the lasso and elastic net. We demonstrate that our approach, called LIT-LVM, achieves superior prediction accuracy compared to the elastic net, hierarchical lasso, and factorization machines on a wide variety of simulated and real data, particularly when the number of interaction terms is high compared to the number of samples. LIT-LVM also provides low-dimensional latent representations for features that are useful for visualizing and analyzing their relationships.

## 1 Introduction

Some of the simplest, yet most commonly used models in statistics and machine learning model an example's target variable $y$ using a weighted linear combination of its features $x_1, x_2, \ldots, x_p$. These linear prediction models, such as linear regression and logistic regression, are favored for their speed, simplicity, and interpretability. In high-dimensional settings where the number of features $p$ is close to or even exceeds the number of examples $n$, regularization techniques such as the lasso (Tibshirani, 1996) and elastic net (Zou & Hastie, 2005), have become essential to prevent overfitting and enhance prediction accuracy on unseen test data.

The primary limitation of linear predictors is their reliance on linear combinations of features, which reduces their flexibility. To enhance their flexibility, a common approach is to incorporate polynomial features, which create new features using products of the original features. Perhaps the simplest case involves computing all second-order *interaction terms* corresponding to products $x_j x_k$ for $j < k$, which describe the interactions between pairs of different features. In classical statistical settings where $p$ is much smaller than $n$, including these $\binom{p}{2}$ interaction terms does not complicate the estimation process, as the total number of features would

---

[*]Research partially conducted while Z. Huang was at Case Western Reserve University.

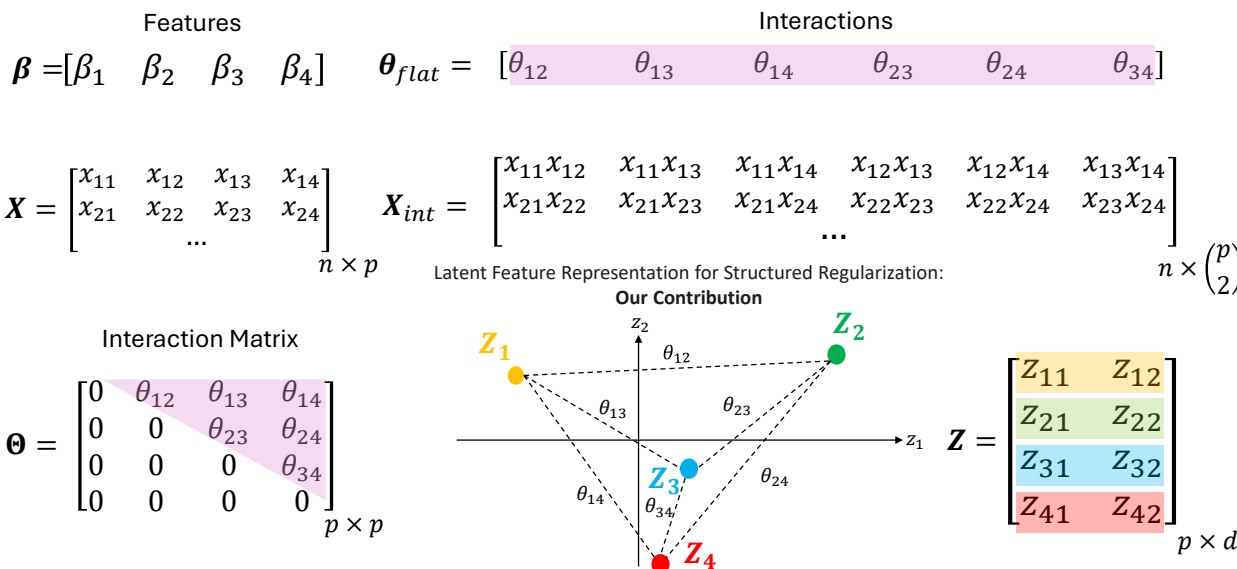

Figure 1: Our proposed framework for linear predictors with interaction terms. We arrange the coefficient vector for the interaction terms $\boldsymbol{\theta}_{\text{flat}}$ into an upper triangular matrix $\boldsymbol{\Theta}$. Our main contribution is to impose an *approximate* low-dimensional structure on $\boldsymbol{\Theta}$ using a structured regularization. We represent each of the $p$ features by a latent vector $\mathbf{z}_j$ in $d$ dimensions, with $d < p$, and minimize the deviation between $\theta_{jk}$ and a function of $\mathbf{z}_j$ and $\mathbf{z}_k$ such as the dot product $\mathbf{z}_j^T \mathbf{z}_k$. In this toy example, $p = 4$ and $d = 2$.

be on the order of $p^2$, which is still quite low compared to $n$. However, as $p$ gets closer to $n$, $p^2$ quickly surpasses $n$, and even stronger regularization may be needed to prevent overfitting.

Unlike the original features, interaction terms have additional structure, as they denote relationships between two features. The coefficients for these interaction terms can be arranged in a $p \times p$ coefficient matrix $\boldsymbol{\Theta}$, where entry $\theta_{jk}$ denotes the coefficient for the interaction term $x_j x_k$. Prior research on structured sparsity for interaction terms has typically assumed a hierarchical structure such that an interaction between features $j, k$ can only be present in the model if either feature $j$ or $k$ is in the model (Bien et al., 2013). We do not make such an assumption and further hypothesize that the coefficient matrix $\boldsymbol{\Theta}$ has an *approximate low-dimensional structure*, so that we can represent each feature $j$ by a vector $\mathbf{z}_j$ in a $d$-dimensional space, where $d < p$.

Our main contribution is a *structured regularization* approach for estimating the coefficients for interaction terms in linear predictors using an approximate low-dimensional structure, as shown in Figure 1, in addition to traditional regularization techniques such as the elastic net. We call our proposed approach LIT-LVM, denoting linear and interaction terms with a latent variable model, and is applicable to many types of linear predictors, including linear regression, logistic regression, and the Cox proportional hazards (Cox PH) model (Cox, 1972). LIT-LVM fits in the space of models between elastic net with interactions, which provides sparse but unstructured interaction coefficient estimates; sparse factorization machines (Atarashi et al., 2021), which use exact low-rank factorizations of the interaction coefficients that are too rigid; and non-linear predictors, which are more expressive but not as interpretable as linear predictors with interactions.

Our proposed structured regularization for the interaction coefficients offers two main benefits. First, it improves estimation accuracy for the interaction coefficients, leading to better predictive accuracy, especially as $p$ grows relative to $n$, which we demonstrate in simulation experiments and across many real datasets for both regression and classification settings. Second, it provides low-dimensional latent representations of the features that are useful for visualizing and analyzing their relationships, which we demonstrate in an application for modeling donor-recipient compatibility in kidney transplantation.

## 2 Background

### 2.1 Linear Predictors

Linear predictors comprise some of the simplest models used in statistics and machine learning. They offer the advantage of being far more interpretable than complex non-linear predictors, though sometimes at the cost of reduced prediction accuracy. Linear predictors generate a prediction $\hat{y}$ of the dependent variable or target $y$ as a function of linear combinations of the independent variables or features $x_1, x_2, \ldots, x_p$:

$$\hat{y} = f(\beta_0 + \beta_1 x_1 + \ldots + \beta_p x_p) = f(\boldsymbol{\beta}^T \mathbf{x}), \tag{1}$$

where $\beta_0$ is an intercept term, and $\beta_1, \ldots, \beta_p$ are the coefficients that quantify the contribution of each feature. They are combined to form the coefficient vector $\boldsymbol{\beta} = [\beta_0, \beta_1, \ldots, \beta_p]$, and the features are combined with a dummy variable of 1 for the intercept to form the feature vector $\mathbf{x} = [1, x_1, \ldots, x_p]$.

**Estimation** In scenarios where the number of examples $n$ significantly exceeds the number of features $p$, the coefficients or weights $\boldsymbol{\beta}$ can typically be accurately estimated using the maximum likelihood estimate (MLE). However, in many application settings, particularly within the biomedical domain, $n$ is often on the same order as $p$ or even smaller, leading to a situation where the MLE can result in severe overfitting. Such overfitting results in models that perform well on training data but poorly on unseen test data. To counteract this, regularization methods like the lasso (Tibshirani, 1996), which uses an $\ell_1$ penalty, and elastic net (Zou & Hastie, 2005), which uses both $\ell_1$ and $\ell_2$ penalties, are often employed to reduce overfitting and enhance the generalization ability of the model. These regularizers tend to favor estimated coefficient vectors that are sparse and have smaller magnitudes.

**Interaction Terms** To capture non-linear relationships between features, linear predictors can be extended by incorporating *interaction terms*. Adding all second-order interaction terms involves adding all products $x_j x_k$ with $j < k$ as additional features. The coefficients for these $\binom{p}{2}$ interaction terms are stored in the *interaction matrix* $\boldsymbol{\Theta}$:

$$\boldsymbol{\Theta} = \begin{bmatrix} 0 & \theta_{12} & \theta_{13} & \cdots & \theta_{1,p} \\ 0 & 0 & \theta_{23} & \cdots & \theta_{2,p} \\ \vdots & \vdots & \vdots & \ddots & \vdots \\ 0 & 0 & 0 & \cdots & \theta_{p-1,p} \\ 0 & 0 & 0 & \cdots & 0 \end{bmatrix}. \tag{2}$$

Let the vectors $\boldsymbol{\theta}_{\text{flat}} = [\theta_{12}, \theta_{13}, \ldots, \theta_{1,p}, \ldots, \theta_{p-1,p}]$ and $\mathbf{x}_{\text{int}} = [x_1 x_2, x_1 x_3, \ldots, x_1 x_p, \ldots, x_{p-1} x_p]$ denote the flattened vector representations of $\boldsymbol{\Theta}$ and the interaction terms. Then, equation 1 can be written as

$$\hat{y} = f(\boldsymbol{\beta}^T \mathbf{x} + \boldsymbol{\theta}_{\text{flat}}^T \mathbf{x}_{\text{int}}) = f(\tilde{\boldsymbol{\beta}}^T \tilde{\mathbf{x}}), \tag{3}$$

where $\tilde{\boldsymbol{\beta}} = [\boldsymbol{\beta}, \boldsymbol{\theta}_{\text{flat}}]$ and $\tilde{\mathbf{x}} = [\mathbf{x}, \mathbf{x}_{\text{int}}]$ denote the augmented coefficient and feature vectors, respectively.

With the addition of interaction terms, the size of the augmented coefficient vector is $\tilde{\boldsymbol{\beta}}$ on the order of $p^2$, so regularization is even more important to prevent overfitting. Since interaction terms represent relations between different features, their coefficients $\boldsymbol{\Theta}$ may possess some additional structure that is not present in the coefficients $\boldsymbol{\beta}$ for the raw features. In this paper, we propose an approximate low-dimensional structure for $\boldsymbol{\Theta}$ that leads to *structured regularization* for interaction coefficients.

### 2.2 Related Work

#### 2.2.1 Interaction Terms and Interpretability

Linear predictors with interaction terms have long been used as an interpretable approach for modeling non-linear relationships between features. Interaction terms can provide the same benefit for generalized additive models (GAMs), as noted by Lou et al. (2013), who proposed the GA$^2$M model to extend GAMs

with pairwise feature interactions, significantly enhancing model accuracy while maintaining interpretability. Higher-order interaction terms can also be added, e.g., in the scalable polynomial additive models (SPAMs) proposed by Dubey et al. (2022), where the maximum order of the interaction terms controls the flexibility of the model. Limiting the maximum order controls the trade-off between flexibility and interpretability, with second-order or pairwise interactions typically representing the "sweet spot" between the two (Dubey et al., 2022).

One challenge when adding interaction terms to linear predictors and GAMs is how to handle a large number of interaction terms. If we add all pairwise interactions to a model, the $p$ features produce $\binom{p}{2}$ interaction terms, which is often a larger number than the number of samples $n$, and can lead to overfitting even when using lasso or elastic net penalties. One approach to avoid this is interaction detection: selectively add only the interactions that pass some statistical test. Lou et al. (2013) adopt this approach and propose a computationally efficient method to rank interactions. Their approach scales to $p$ in the thousands, which results in millions of possible pairwise interactions. Another approach is to assume that the interaction matrix $\Theta$ possesses some low-dimensional structure, such as low rank. This is the assumption made by factorization machines, which we discuss in Section 2.2.3.

Another line of research on interaction terms focuses on structured sparsity using the notion of heredity. Under the strong heredity structure, the interaction coefficient $\theta_{jk} \neq 0$ only if $\beta_j \neq 0$ and $\beta_k \neq 0$, implying that both the features $j, k$ must be selected in order to select their interaction. Several regularization methods that incorporate strong heredity or a relaxed version termed weak heredity have been proposed (Zhao et al., 2009; Yuan et al., 2009; Choi et al., 2010; Radchenko & James, 2010; Bien et al., 2013). The hierarchical lasso (Bien et al., 2013) is probably the most similar to the methods we consider in this paper, as it also uses an elastic net penalty, where the $\ell_1$ portion enforces the strong or weak heredity structure. The heredity assumption simplifies the discovery of interaction terms, as one can first select a smaller number of features through their main effects $\boldsymbol{\beta}$ and then consider only interactions between the selected terms, but does not assume any low-dimensional structure on the interaction matrix.

### 2.2.2 Latent Variable Models

Latent variable models (LVMs) are used extensively in statistics and machine learning and aim to uncover latent variables representing hidden factors not directly observable but inferred from available data. An LVM particularly relevant to this paper is the latent space model (LSM) proposed by Hoff et al. (2002) for graph-structured data. Under this model, conditional independence of node pairs is assumed, given the latent positions of the nodes in a Euclidean space. In the most commonly used form of the LSM, the log odds of an edge forming between two nodes $i$ and $j$ is given by $\alpha_0 + \alpha_1 x_{ij} - \|\mathbf{z}_i - \mathbf{z}_j\|_2$, where $x_{ij}$ denotes observed covariates about the node pair, $\mathbf{z}_i$ and $\mathbf{z}_j$ are the latent positions of nodes $i$ and $j$ in a $d$-dimensional latent space, and $\alpha_0$ and $\alpha_1$ are scalar parameters. This model implies a greater probability of an edge between nodes that are closer together in the latent space.

Huang & Xu (2022) proposed a novel LSM to analyze human leukocyte antigen (HLA) compatibility from data on kidney transplant outcomes. They first fit a Cox PH model with interaction terms using an $\ell_2$ penalty. They propose to improve the estimates of the interaction coefficients using an LSM for the HLA compatibility network. Their proposed approach uses two-step estimation: first optimizing the $\ell_2$-penalized Cox PH model's partial likelihood to estimate interaction coefficients and then optimizing the LSM likelihood to refine the estimates. On the other hand, our approach uses joint estimation over the model coefficients and latent vectors and can be applied broadly to many linear predictors and regularizers.

### 2.2.3 Factorization Machines (FMs) and Related Models

The first FM model was proposed by Rendle (2010). It assumes that each feature $j$ has a $d$-dimensional latent vector $\mathbf{z}_j$ such that $\Theta = \mathbf{Z}\mathbf{Z}^T$. This forces $\Theta$ to have an *exact* low-rank structure. Sparse FMs (Xu et al., 2016; Pan et al., 2016; Atarashi et al., 2021) further constrain $\Theta$ to be sparse; we discuss these models further in Section 3.3.

Factorization of interaction terms has also been applied to models with time-varying coefficients through the factorized structured regression (FaStR) model proposed by Rügamer et al. (2022). FMs have been used for

modeling in recommender systems, and the extension to time-varying coefficients in FaStR enables modeling in time-aware recommender systems where user behaviors change over time. FMs have also been extended to higher-order interaction terms in a scalable manner, including higher-order FMs (Blondel et al., 2016) and additive higher-order FMs (Rügamer, 2024), which combine the benefits of GAMs with the scalability of factorized terms.

FMs and other models with factorized coefficients differ from our proposed LIT-LVM approach in two main aspects. First, they are heavily concerned with scalability when using interaction terms and are often applied to data with both large $n$ and $p$ with a sparse feature matrix $\mathbf{X}$ (Rendle, 2012; 2013). On the other hand, our focus is on improving accuracy of estimation and prediction, not scalability, so our approach is not more scalable than directly modeling interaction terms. Second, FMs and other models with factorized coefficients assume that $\boldsymbol{\Theta}$ is *exactly* low rank, whereas we assume that $\boldsymbol{\Theta}$ has an *approximate* low-dimensional structure (possibly low rank, but not necessarily). This is a subtle, yet important distinction, and we provide more details in Section 3.3.

# 3 Model Formulation and Estimation

## 3.1 LVM for Interaction Terms

We propose to improve the estimation accuracy of the coefficients for interaction terms by introducing an LVM for these coefficients. We make an underlying assumption that the $p \times p$ interaction coefficient matrix $\boldsymbol{\Theta}$ has an *approximate* low-dimensional structure captured by the matrix $p \times d$ matrix $\mathbf{Z}$. (Note that this does not imply a low-dimensional structure for the $n \times p$ data matrix $\mathbf{X}$ itself.) We discuss two possible models in the following, a low rank and a latent distance model. Other continuous LVMs or matrix factorizations could also be used instead, such as non-negative matrix factorization (Lee & Seung, 1999).

**Low Rank Model** This model hypothesizes that the interaction matrix $\boldsymbol{\Theta}$ may exhibit an approximate low rank structure. Under this model, the interaction coefficient for any pair $j, k$ such that $j < k$ is given by

$$\theta_{jk} = \mathbf{z}_j^T \mathbf{z}_k + \epsilon_{jk}, \tag{4}$$

where $\epsilon_{jk}$ is a zero-mean error term for the pair $j, k$ representing the deviation from the low rank structure. This model is closely related to factorization machines, which assume that $\theta_{jk} = \mathbf{z}_j^T \mathbf{z}_k$, i.e., an exact rather than approximate low rank structure.

**Latent Distance Model** This model is inspired by the LSM of Hoff et al. (2002), where the distances between latent vectors is of interest. Under this model, the interaction coefficient for any pair $j, k$ such that $j < k$ is defined as

$$\theta_{jk} = \alpha_0 - \|\mathbf{z}_j - \mathbf{z}_k\|_2^2 + \epsilon_{jk}, \tag{5}$$

where the intercept $\alpha_0$ denotes the baseline interaction level. The latent distance model has a negative effect for distance so that features with more positive values of interactions are placed closer together in the latent space and more negative values of interactions are pushed farther apart. This type of negative effect on distance has also been used for matrices with both positive and negative entries by Huang & Xu (2022) and Nakis et al. (2023) and is often more interpretable than models with a positive effect on distance (Huang et al., 2022).

## 3.2 Estimation Procedure

We propose to estimate the model parameters $\tilde{\boldsymbol{\beta}}$, which denotes the augmented coefficient vector combining $\boldsymbol{\beta}$ and the flattened version of $\boldsymbol{\Theta}$ as described in Section 2.1, by minimizing the total loss function

$$\mathcal{L}_{\text{total}} = \mathcal{L}_{\text{pred}} + \lambda_r \mathcal{L}_{\text{reg}} + \lambda_l \mathcal{L}_{\text{lvm}}, \tag{6}$$

where $\lambda_r$ and $\lambda_l$ are hyperparameters controlling the strength of the traditional regularization and the LVM-structured regularization, respectively.

The initial term, $\mathcal{L}_{\text{pred}}$, addresses the conventional loss associated with the linear predictor, such as mean-squared error for linear regression, log loss for logistic regression, and negative log partial likelihood for the Cox PH model. The second component, $\mathcal{L}_{\text{reg}}$, is associated with standard regularization techniques for linear predictors, such as the elastic net (Zou & Hastie, 2005). The third component, $\mathcal{L}_{\text{lvm}}$, is associated with the LVM for the interaction term weights. This component ensures that the estimated interaction coefficients $\hat{\boldsymbol{\Theta}}$ align with the hypothesized approximate low-dimensional structure.

Depending on the choice of linear predictor, regularizer, and LVM, the form of the total loss function will differ. For example, for the case of linear regression with elastic net regularization and a low rank LVM, the total loss would be

$$\mathcal{L}_{\text{total}}(\boldsymbol{\beta}, \boldsymbol{\Theta}, \mathbf{Z}) = \underbrace{\frac{1}{n}\|\mathbf{y} - \tilde{\mathbf{X}}\tilde{\boldsymbol{\beta}}\|_2^2}_{\text{mean squared error}} + \underbrace{\lambda_2\|\tilde{\boldsymbol{\beta}}\|_2^2 + \lambda_1\|\tilde{\boldsymbol{\beta}}\|_1}_{\text{elastic net regularizer}} + \underbrace{\lambda_l\|\boldsymbol{\Theta} - \mathbf{Z}\mathbf{Z}^T\|_F^2}_{\text{low rank LVM}}.$$

**Our Contribution**   Our main contribution is the third term $\mathcal{L}_{\text{lvm}} = \|\boldsymbol{\epsilon}\|_F^2$ in equation 6, denoting the deviation from a low-dimensional structure. $\boldsymbol{\epsilon}$ is the matrix of error terms between the interaction coefficients $\theta_{jk}$ and reconstructions from their latent representations $\mathbf{z}_j, \mathbf{z}_k$. By minimizing squared distances between the interaction coefficients and their reconstructions from low-dimensional representations, we encourage the interaction coefficients to have an *approximate* low-dimensional structure. Our structured regularization approach can be easily modified for settings where not all interactions are of interest, as we describe in Section A.1.

**Optimization**   We use Adam (Kingma & Ba, 2015) with various learning rates $\in [0.005, 0.01, 0.05, 0.1]$ as our optimizer for the total loss function. To handle the $\ell_1$ penalty, we use the proximal gradient method (Parikh & Boyd, 2014), where the proximal operator corresponds to the soft thresholding operator. The other terms are all differentiable, with derivatives computed by automatic differentiation in PyTorch. The initializations of $\tilde{\boldsymbol{\beta}}$ and $\mathbf{Z}$ are drawn from normal distributions: $\tilde{\beta}_j \sim \mathcal{N}(0, 1)$ and $z_{jk} \sim \mathcal{N}(0, 1)$. The time complexity of our estimation procedure per epoch is $O(np^2)$, as we discuss in Section A.2.

### 3.3 Comparison with Related Methods

**Nuclear Norm Regularization**   One approach to estimating low rank and sparse matrices is by adding nuclear norm and $\ell_1$ penalties. Such an approach has been used in robust principal component analysis (PCA) (Candès et al., 2011) to estimate a matrix of the form $\mathbf{M} = \mathbf{L}_0 + \mathbf{S}_0$, where $\mathbf{L}_0$ is low rank and $\mathbf{S}_0$ is sparse.

More closely related to our work, Richard et al. (2012) and Zhou et al. (2013) propose approaches also using nuclear norm and $\ell_1$ penalties to estimate a single matrix $\mathbf{S}$ that is simultaneously sparse and low rank. If we applied these penalties to linear regression with interaction terms, it would result in the loss function

$$\frac{1}{n}\|\mathbf{y} - \tilde{\mathbf{X}}\tilde{\boldsymbol{\beta}}\|_2^2 + \lambda_1\|\tilde{\boldsymbol{\beta}}\|_1 + \lambda_l\|\boldsymbol{\Theta}\|_*,$$

where $\|\boldsymbol{\Theta}\|_* = \sum_j \sigma_j(\boldsymbol{\Theta})$ denotes the nuclear or trace norm consisting of the sum of the singular values of $\boldsymbol{\Theta}$. This approach has the advantage of being convex; however, the disadvantage is that it requires repeated computation of a singular value decomposition as part of the optimization routine, which has $O(p^3)$ complexity. If we adopted this approach for our setting rather than introducing the latent variables $\mathbf{Z}$, then time complexity per epoch would be $O(np^3)$, higher than the $O(np^2)$ of our LIT-LVM approach. Additionally, we are also interested in the latent vectors $\mathbf{z}_j$ themselves, which can also help to interpret the structure of the interaction matrix $\boldsymbol{\Theta}$.

**Sparse Factorization Machines**   Another approach to estimating low rank and sparse matrices is sparse FMs, which have been developed to improve the performance and interpretability of FMs (Xu et al., 2016; Pan et al., 2016; Atarashi et al., 2021). Sparse FMs model the interaction coefficients by $\boldsymbol{\Theta} = \mathbf{Z}\mathbf{Z}^T$ such that $\boldsymbol{\Theta}$ is sparse. Since $\mathbf{Z}$ is $p \times d$ with $d < p$, sparse FMs constrain $\boldsymbol{\Theta}$ to have at most rank $d$, i.e., an exact low-rank representation. On the other hand, our proposed LIT-LVM approach, when used with a low rank

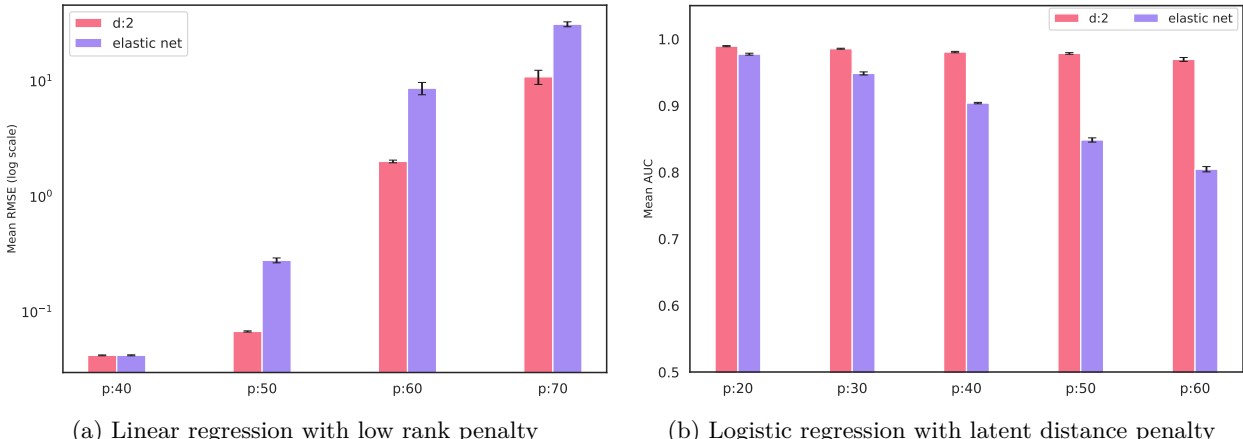

(a) Linear regression with low rank penalty  (b) Logistic regression with latent distance penalty

Figure 2: Results of simulation experiments on linear and logistic regression where the true latent dimension for the interaction weights $\boldsymbol{\Theta}$ is $d_{\text{true}} = 2$. (a) Lower RMSE and (b) higher AUC denote higher prediction accuracy. Error bars denote one standard error. Our proposed LIT-LVM approach, which uses structured regularization, significantly outperforms the elastic net with interaction terms, a traditional regularization approach, as $p$ increases for fixed $n$.

model, penalizes the deviation from a low rank representation using the penalty $\|\boldsymbol{\Theta} - \mathbf{Z}\mathbf{Z}^T\|_F^2$, resulting in $\boldsymbol{\Theta}$ being *approximately* rather than exactly low rank and enabling greater modeling flexibility. The trade-off for the greater flexibility is higher time and space complexity as we need to work with the entire $p \times p$ matrix $\boldsymbol{\Theta}$ rather than just with the $p \times d$ matrix $\mathbf{Z}$. In the limit as $\lambda_l \to \infty$, the LIT-LVM estimate of $\boldsymbol{\Theta}$ approaches that of a sparse FM, as any deviation from the low rank structure is heavily penalized.

## 4 Simulation Experiments

We begin with some simulation experiments to investigate the effects of LIT-LVM as we change the number of features $p$ with fixed number of examples $n$. As $p$ increases, the model becomes more difficult to estimate. Code to reproduce all experiments is available at `https://github.com/MLNS-Lab/LIT-LVM`.

### 4.1 Linear Regression with Low Rank

We construct a feature matrix $\mathbf{X}$ with dimensions $n \times p$. Each data point $\mathbf{x}_i \sim \mathcal{N}(\mathbf{0}, \mathbf{I})$. From the feature matrix, we then construct the matrix of interaction terms $\mathbf{X}_{\text{int}}$, which has dimensions $n \times \binom{p}{2}$. The target vector $\mathbf{y}$ is then generated by $\mathbf{y} = \mathbf{X}\boldsymbol{\beta} + \mathbf{X}_{\text{int}}\boldsymbol{\theta}_{\text{flat}} + \boldsymbol{\eta}$. The entries of the weight vector $\boldsymbol{\beta}_j \sim \mathcal{N}(0, 1)$. $\mathbf{Z}$ has dimensions $p \times d_{\text{true}}$ with $d_{\text{true}} < p$ so that $\boldsymbol{\Theta}$ possesses the approximate low rank structure in equation 4, with entries $z_{jk} \sim \mathcal{N}(0, 1)$ and $\epsilon_{jk} \sim \mathcal{N}(0, 0.1)$. Finally, the observation noise $\eta_i \sim \mathcal{N}(0, 0.01)$.

**Results** The results for fixed $n = 1{,}000$, true latent dimension $d_{\text{true}} = 2$, and four different values of $p$ are shown in Figure 2a. We compare our proposed LIT-LVM approach using a low rank model to elastic net with interaction terms. For $p = 40$, $\boldsymbol{\Theta}$ has $\binom{40}{2} = 780$ entries, which is less than $n$, and both approaches are quite accurate. At $p = 50$, $\boldsymbol{\Theta}$ has $\binom{50}{2} = 1{,}225$ entries, which now exceeds $n$, and we observe a phase transition where the RMSE of elastic net increases significantly compared to our LIT-LVM approach. As $p$ continues to increase, LIT-LVM continues to outperform the elastic net with interactions.

In Section B.1, we repeat the experiment for true latent dimensions of 5 and 10 and observe similar results. We also evaluate LIT-LVM for $d \neq d_{\text{true}}$ and find that, although the accuracy drops, it drops less when choosing $d$ too high rather than too low. Furthermore, its accuracy is still higher than elastic net even with misspecified $d$.

## 4.2 Logistic Regression with Latent Distance

In this experiment, we generate the feature matrix and interaction terms in the same manner as in the linear regression experiment in Section 4.1. For logistic regression, each element of the response vector $\mathbf{y}$ is sampled from a Bernoulli distribution with success probability given by $\text{logistic}(\boldsymbol{\beta}^T\mathbf{x} + \boldsymbol{\theta}_{\text{flat}}^T\mathbf{x}_{\text{int}}) + \eta_i$, where $\text{logistic}(x) = \frac{1}{1+e^{-x}}$. We inject *sparsity* into the augmented coefficient vector $\tilde{\boldsymbol{\beta}}$ by first generating a dense version $\tilde{\boldsymbol{\beta}}^{\text{dense}}$. We then set each element $\tilde{\beta}_j^{\text{dense}}$ to 0 with probability given by the Gaussian function $\exp(-(\tilde{\beta}_j^{\text{dense}})^2/\sigma_s^2)$, where $\sigma_s^2$ is a parameter that adjusts the level of sparsity. This ensures that larger elements in $\tilde{\boldsymbol{\beta}}^{\text{dense}}$ have a lower probability of being zeroed out.

To generate the dense augmented coefficient vector $\tilde{\boldsymbol{\beta}}^{\text{dense}} = [\boldsymbol{\beta}^{\text{dense}}, \boldsymbol{\theta}_{\text{flat}}^{\text{dense}}]$, we sample entries for the weight vector $\boldsymbol{\beta}^{\text{dense}}$ from $\beta_j^{\text{dense}} \sim \mathcal{N}(0,1)$. The interaction matrix $\boldsymbol{\Theta}^{\text{dense}}$ is constructed using the latent distance model in equation 5. The matrix $\boldsymbol{\epsilon}$ with entries $\epsilon_{jk} \sim \mathcal{N}(0,\sigma_\theta^2)$ denotes the deviation from the latent distance model's assumed structure. This formulation captures the interaction through the distances in the latent space between each feature, where the matrix $\boldsymbol{\Theta}^{\text{dense}}$ is flattened into $\boldsymbol{\theta}_{\text{flat}}^{\text{dense}}$ before it is sparsified. We also model noise in the response variable using a Gaussian distribution such that $\eta_i \sim \mathcal{N}(0,\sigma_y^2)$. We set the variances of the Gaussian distributions to $\sigma_s^2 = 0.0001$, $\sigma_\theta^2 = 0.1$, and $\sigma_y^2 = 0.01$.

**Results**  The results for fixed $n = 1,000$, true latent dimension $d_{\text{true}} = 2$, and five different values of $p$ are shown in Figure 2b. Similar to the results from the linear regression experiment, we observe a phase transition where the accuracy of elastic net decreases significantly compared to our approach. The phase transition occurs at roughly $p = 30$, lower than in the linear regression experiment. As $p$ continues to increase, our LIT-LVM approach continues to outperform elastic net, with prediction accuracy decreasing much slower than in elastic net.

In Section B.2, we repeat the experiment for true latent dimension of 5 and 10 and observe similar results. We also evaluate the performance of LIT-LVM for misspecified $d$, with the same findings as for linear regression: choosing $d$ too high is better than too low. We also demonstrate superior performance of LIT-LVM over elastic net persists in even higher-dimensional settings up to $p = 500$.

**Comparison with Sparse FMs**  In Section B.3, we conduct simulation experiments comparing the performance of LIT-LVM with sparse FMs[1]. We simulate data in the same manner except with a low rank structure rather than a latent space structure, to better match the assumption of FMs. Likewise, we use LIT-LVM with a low rank model. We find that FMs perform worse when $\boldsymbol{\Theta}$ is noisy or exhibits higher sparsity (deviation from exact low rank). This indicates that factorizing the interaction coefficients, as FMs do, can degrade performance when the underlying assumption of exact low rank structure is not met. In contrast, the approximate low rank structure assumption in LIT-LVM offers greater flexibility in modeling the interaction coefficients, ensuring more robustness and higher accuracy.

**Effects of LVM Hyperparameter** $\lambda_l$  Next, we investigate the sensitivity of LIT-LVM to the hyperparameter $\lambda_l$, which controls the strength of the LVM-structured regularization. On one extreme, $\lambda_l = 0$ reduces LIT-LVM to a standard regularized linear predictor, while very large $\lambda_l$ enforces a strict low-dimensional structure that approaches factorization machines as $\lambda_l \to \infty$.

First, we consider the simulation with true latent dimension $d_{\text{true}} = 2$ and low noise in the latent variables, where $\epsilon_{jk}$, the deviation from low rank structure, has variance $\sigma_\theta^2 = 0.1$, so that the interaction matrix remains approximately low rank. Figure 3a shows the mean AUC for varying $\lambda_l$ while holding the other regularization hyperparameters ($\lambda_1, \lambda_2$) fixed. When $\lambda_l = 0$, the model can only learn a purely sparse linear predictor with no latent structure penalty, which leads to relatively low predictive accuracy. On the other hand, at extremely high $\lambda_l$, LIT-LVM approaches a factorization machine, which also underperforms here because it enforces an overly strict *exact* low-rank assumption. Notably, there is a middle range of $\lambda_l$ that yields the highest mean AUC, indicating that partially constraining interaction terms to follow a latent structure can improve prediction while allowing some flexibility for deviations.

---

[1]To avoid differences in performance caused by the optimizer, we implement sparse FMs using LIT-LVM with $\lambda_l = 100,000$. We show in Section C.3 that this value is sufficient. (Recall that LIT-LVM approaches a sparse FM in the limit as $\lambda_l \to \infty$.).

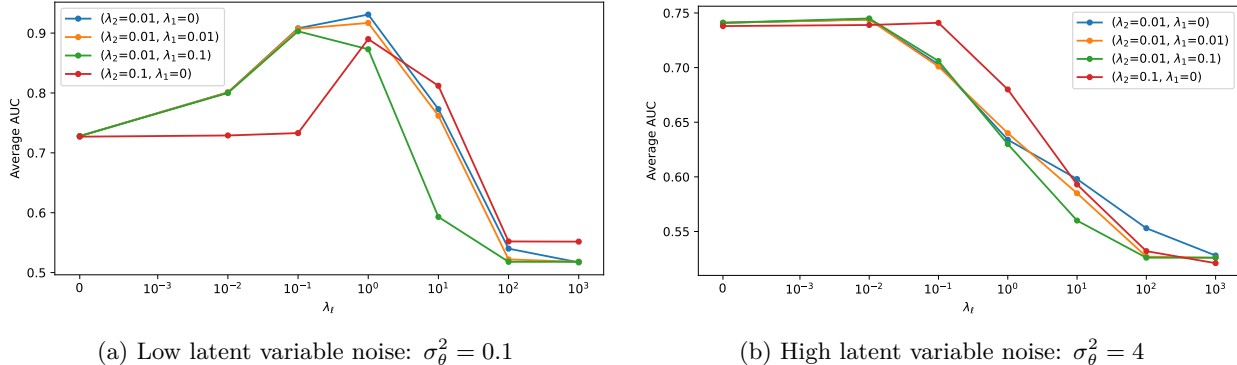

(a) Low latent variable noise: $\sigma_\theta^2 = 0.1$      (b) High latent variable noise: $\sigma_\theta^2 = 4$

Figure 3: Mean AUC for different values of the LVM regularization strength $\lambda_l$ on simulated data experiments using logistic regression. (a) The simulation with low noise ($\sigma_\theta^2 = 0.1$) peaks in AUC at an intermediate $\lambda_l$, confirming the benefit of the LVM regularization term. (b) The simulation with high noise ($\sigma_\theta^2 = 4$) has destroyed the latent structure, and thus shows minimal benefit from the LVM regularization.

To assess the robustness of LIT-LVM under severe violations of the low-rank assumption, we repeat the same simulation but increase latent variable noise variance to $\sigma_\theta^2 = 4$, effectively destroying the underlying low-rank structure. As shown in Figure 3b, the best performing $\lambda_l$ is near 0 or 0.01 which corresponds to applying no or little LVM penalty. This is not surprising because at such high noise, the interaction matrix no longer possesses a low-rank structure so larger values of $\lambda_l$ degrade the performance.

## 5  Real Data Experiments

### 5.1  OpenML Datasets

We conduct experiments on a wide variety of publicly available datasets from OpenML (Vanschoren et al., 2013) to verify how well our structured regularization for the interaction matrix applies to real data. We consider both regression and binary classification tasks using linear and logistic regression, respectively. We consider 12 regression datasets and 10 classification datasets, described in Section C.1. For each task and dataset, we compare the accuracy of five different predictors, all with an elastic net penalty, listed below in increasing order of flexibility:

1. Elastic net with no interactions: the base linear predictor (linear or logistic regression) applied directly to the features.

2. Sparse factorization machines (FMs): factorized form for the interaction coefficient matrix that assumes exact low rank structure.

3. LIT-LVM: our proposed approach that assumes approximate low rank structure for the interaction coefficient matrix.

4. Hierarchical lasso: includes only interaction terms $x_j x_k$ where at least one of the features (main effects) $x_j$ or $x_k$ is also included. This corresponds to an assumption of weak heredity.

5. Elastic net with interactions: includes interaction terms between all pairs of features.

We divide the data into a 50/50 train/test split for both classification and regression tasks. We tune hyperparameters using a grid search, selecting the set of hyperparameters that minimize the mean RMSE for regression tasks and maximize the mean AUC for classification tasks over five splits within the training set. Subsequently, we evaluate prediction accuracy on the held-out test set. We repeat this process using five different train/test splits to calculate the standard errors for the prediction accuracy. Details on hyperparameter tuning are provided in Section C.1.2.

Table 1: Mean RMSE ± standard error over 5 different train/test splits for OpenML regression datasets. Lower RMSE denotes higher prediction accuracy. Highest accuracy for each data set is denoted in **bold**, and entries within 1 standard error are denoted in *italic*. Our proposed LIT-LVM approach has the highest accuracy on 11 of the 12 datasets and is within 1 standard error on the 1 remaining dataset.

| Dataset Name | Elastic Net No Interactions | Elastic Net with Interactions | LIT-LVM | Sparse FM | Hierarchical Lasso |
|---|---|---|---|---|---|
| boston | $4.99 \pm 0.12$ | $\mathbf{4.39 \pm 0.23}$ | *4.40 ± 0.26* | *5.01 ± 0.63* | *4.51 ± 0.17* |
| bike_sharing_demand | $148.40 \pm 1.08$ | *144.48 ± 1.06* | $\mathbf{144.47 \pm 1.05}$ | $156.65 \pm 1.07$ | *144.56 ± 0.39* |
| diamonds | $0.28 \pm 0.00$ | $0.26 \pm 0.00$ | $\mathbf{0.25 \pm 0.00}$ | $0.28 \pm 0.00$ | $0.28 \pm 0.00$ |
| fri_c4_500_10 | *0.88 ± 0.01* | *0.89 ± 0.02* | $\mathbf{0.87 \pm 0.02}$ | $1.00 \pm 0.03$ | *0.88 ± 0.02* |
| medical_charges | $0.24 \pm 0.00$ | $\mathbf{0.19 \pm 0.00}$ | $0.19 \pm 0.00$ | $0.20 \pm 0.00$ | $0.20 \pm 0.00$ |
| nyc_taxi_green | $0.50 \pm 0.00$ | $0.50 \pm 0.00$ | $\mathbf{0.16 \pm 0.00}$ | $0.50 \pm 0.00$ | $0.27 \pm 0.00$ |
| pol | $30.55 \pm 0.08$ | *24.33 ± 0.71* | $\mathbf{24.24 \pm 0.63}$ | $29.05 \pm 0.33$ | $28.41 \pm 0.06$ |
| rmftsa_ladata | $1.86 \pm 0.05$ | *1.73 ± 0.05* | $\mathbf{1.72 \pm 0.03}$ | $1.85 \pm 0.05$ | $6.49 \pm 0.08$ |
| socmob | $23.57 \pm 0.61$ | *17.17 ± 1.15* | $\mathbf{16.55 \pm 1.56}$ | $23.56 \pm 1.96$ | *17.52 ± 1.05* |
| space_ga | $0.14 \pm 0.00$ | $0.13 \pm 0.00$ | $\mathbf{0.12 \pm 0.00}$ | $0.13 \pm 0.00$ | $0.14 \pm 0.00$ |
| superconduct | $17.72 \pm 0.07$ | $15.17 \pm 0.19$ | $\mathbf{14.32 \pm 0.07}$ | $17.70 \pm 0.08$ | $18.42 \pm 0.05$ |
| wind | $3.30 \pm 0.01$ | $3.28 \pm 0.03$ | $\mathbf{3.17 \pm 0.02}$ | $3.28 \pm 0.01$ | $3.47 \pm 0.02$ |

Table 2: Mean AUC ± standard error over 5 different train/test splits for OpenML classification datasets. Higher AUC denotes higher prediction accuracy. Highest accuracy for each data set is denoted in **bold**, and entries within 1 standard error are denoted in *italic*. Our proposed LIT-LVM approach has the highest accuracy on 9 of the 10 datasets and is within 1 standard error on the 1 remaining dataset.

| Dataset Name | Elastic Net No Interactions | Elastic Net with Interactions | LIT-LVM | Sparse FM | Hierarchical Lasso |
|---|---|---|---|---|---|
| Bioresponse | $0.791 \pm 0.013$ | $0.791 \pm 0.004$ | $\mathbf{0.808 \pm 0.003}$ | *0.801 ± 0.004* | $\mathbf{0.808 \pm 0.004}$ |
| clean1 | $0.900 \pm 0.003$ | $0.922 \pm 0.004$ | $\mathbf{0.947 \pm 0.002}$ | $0.900 \pm 0.005$ | $0.934 \pm 0.003$ |
| clean2 | $0.944 \pm 0.001$ | $0.943 \pm 0.003$ | $\mathbf{0.985 \pm 0.001}$ | $0.955 \pm 0.001$ | $0.978 \pm 0.001$ |
| eye_movements | $0.586 \pm 0.004$ | $\mathbf{0.608 \pm 0.002}$ | *0.607 ± 0.002* | $0.586 \pm 0.004$ | *0.604 ± 0.005* |
| fri_c4_500_100 | $\mathbf{0.666 \pm 0.011}$ | $0.576 \pm 0.020$ | $\mathbf{0.666 \pm 0.010}$ | *0.651 ± 0.012* | $\mathbf{0.666 \pm 0.010}$ |
| fri_c4_1000_100 | *0.672 ± 0.009* | $0.564 \pm 0.009$ | $\mathbf{0.689 \pm 0.008}$ | *0.674 ± 0.014* | *0.678 ± 0.009* |
| hill-valley | *0.711 ± 0.020* | $0.547 \pm 0.027$ | $\mathbf{0.734 \pm 0.034}$ | *0.699 ± 0.023* | *0.715 ± 0.020* |
| jannis | $0.814 \pm 0.000$ | *0.826 ± 0.001* | $\mathbf{0.827 \pm 0.001}$ | $0.801 \pm 0.001$ | $0.821 \pm 0.001$ |
| jasmine | *0.831 ± 0.004* | *0.836 ± 0.003* | $\mathbf{0.837 \pm 0.003}$ | *0.835 ± 0.005* | *0.832 ± 0.003* |
| tecator | *0.969 ± 0.004* | $0.959 \pm 0.006$ | $\mathbf{0.974 \pm 0.003}$ | *0.970 ± 0.004* | *0.970 ± 0.003* |

**Results** The regression and classification results are shown in Tables 1 and 2, respectively. Our LIT-LVM approach achieves superior accuracy over the suite of datasets for both tasks, and by a large margin on some datasets, such as nyc_taxi_green for regression and clean_2 for classification. As we observed in the simulation experiments, our proposed LIT-LVM approach achieves the highest gains compared to elastic net with interactions on datasets with high $p^2/n$ ratios (shown in Tables 4 and 5 in Section C.1.1), including superconduct for regression and fri_c4_1000_100 for classification. This also agrees with well-known results for other regularization methods such as elastic net, where the benefit is mainly achieved when the ratio $p/n$ approaches or exceeds 1. With the inclusion of $\binom{p}{2}$ interaction terms, it is the ratio $p^2/n$ that is important. We also note that, without the structured regularization of LIT-LVM, interaction terms can often hurt the accuracy of a linear predictor, as shown by classification datasets such as hill-valley, where elastic net without interactions far exceeds the accuracy with interactions. Such results were also observed by Bien et al. (2013) in their experiments comparing elastic net without interactions, with interactions, and hierarchical lasso. We find that the hierarchical lasso's prediction accuracy is generally in between that of elastic net with and without interactions, as one might expect. The weak heredity assumption provides a stronger regularization compared to elastic net with interactions, which allows hierarchical lasso to perform

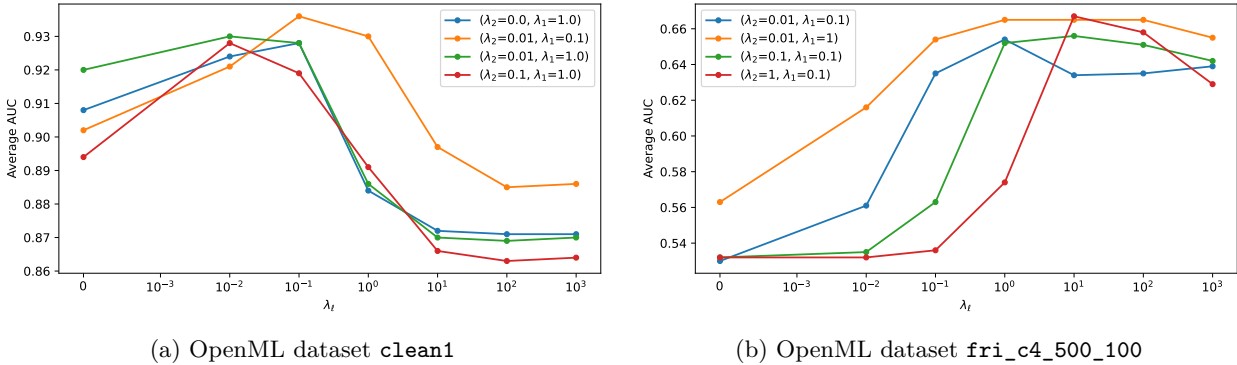

(a) OpenML dataset `clean1`

(b) OpenML dataset `fri_c4_500_100`

Figure 4: Mean AUC for different values of the LVM regularization strength $\lambda_l$ on real OpenML classification datasets. (a) The `clean1` dataset peaks in AUC at a moderate $\lambda_l$ between 0.01 and 0.1, while the more difficult `fri_c4_500_100` dataset benefits from stronger LVM regularization with $\lambda_l$ in the range of 1 to 100.

better, but not as well as LIT-LVM, on datasets with high $p^2/n$ ratios. On all datasets, we find that our proposed optimization procedure for LIT-LVM converges, as shown in Section C.2.

Additionally, we find that FMs are not competitive with our proposed LIT-LVM approach in accuracy, despite having higher latent dimension up to $d = 50$ tuned by grid search, compared to LIT-LVM with fixed $d = 2$. We believe that this is due to its assumption that the interaction terms have an exact low rank structure, rather than an approximate one as in LIT-LVM. A comparison of LIT-LVM and FM for varying $d$ is provided in Section C.1.3.

**Effects of LVM Hyperparameter** $\lambda_l$   The effects of $\lambda_l$ on mean AUC in the `clean1` dataset are shown in Figure 4a. From Table 2, we see that LIT-LVM already attains a substantially higher AUC on `clean1` than the other baselines. We observe that moderate values of $\lambda_l$ again yield the best result, reinforcing that some, but not excessive, emphasis on latent structure can exploit the potential low-rank signal in the data. The performance on another data set, `fri_c4_500_100` is shown in Figure 4b. From Table 2, we see that the mean AUC of FM and LIT-LVM are closer on this dataset than on `clean1`, which indicates that LIT-LVM is indeed imposing a low-rank structure similar to FM, but less rigid, and thus, still manages to outperform it. Consequently, we observe a higher value for $\lambda_l$ to maximize AUC compared to `clean1`. Nevertheless, the overall pattern remains similar: a moderate $\lambda_l$ still achieves or exceeds the best possible performance from an FM, which is a purely factorized method, or a model with just an elastic net penalty and no factorization.

## 5.2   Survival Prediction and Compatibility Estimation for Kidney Transplantation

We now turn to a biomedical application that partially motivated the development of our LIT-LVM approach. We aim to predict the outcomes of kidney transplants in terms of the time duration until the transplanted kidney eventually fails. This is known as the time to graft failure or *survival time*. Survival times are influenced by many factors, including the compatibility of the Human Leukocyte Antigens (HLAs) between the donor and recipient (Opelz et al., 1999; Foster et al., 2014). HLA compatibility also impacts the accuracy of survival time predictions (Nemati et al., 2023).

**Data Description**   This study used data from the Scientific Registry of Transplant Recipients (SRTR). The SRTR data system includes data on all donor, wait-listed candidates, and transplant recipients in the US, submitted by the members of the Organ Procurement and Transplantation Network (OPTN). The Health Resources and Services Administration (HRSA), U.S. Department of Health and Human Services provides oversight to the activities of the OPTN and SRTR contractors.

We follow the inclusion criteria and data preprocessing steps outlined by Nemati et al. (2023) on the SRTR kidney transplant data. Each sample includes demographic and clinical information of both donors and recipients, such as age, gender, race, and BMI. Additionally, we incorporate data on the HLA types of donors

and recipients. Since it is the compatibility between the HLAs of the donors and recipients that is the important factor influencing survival times, it is the interactions between features representing donor HLAs and recipient HLAs, called HLA pairs, that are of interest. We refer to these HLA pairs using uppercase letters for the donor and lowercase for the recipient, similar to Huang & Xu (2022), e.g., A1_a2 denotes that the donor has HLA-A1 and the recipient has HLA-A2. We further describe the construction of these interaction terms in Section C.4.

Our dataset incorporates three specific HLA loci, which can be interpreted as groups of features: HLA-A, HLA-B, and HLA-DR. We construct separate interaction matrices for each locus. Furthermore, the interaction matrix considers only interactions between donor and recipient HLA types. This targeted approach enables us to focus on the critical aspect of how donor-recipient HLA interactions impact graft survival time. We describe how our proposed LIT-LVM model can be formulated with such targeted interactions in Section A.

Over 70% of the examples are censored, mostly denoting transplants that have not yet failed. Thus, we utilize the Cox Proportional Hazards (Cox PH) model, a linear predictor commonly used for survival analysis, rather than standard regression models to account for the censored examples. We divide the transplant data into training and test sets, with a 50/50 split. We select the set of hyperparameters that maximize the mean Cox PH partial log-likelihood using a 5-fold cross-validation on the training set. We evaluate prediction accuracy on the test set using Harrell's concordance index (C-index) (Harrell et al., 1982), which is a ranking-based metric that measures only discrimination ability, and integrated time-dependent Brier score (IBS) (Graf et al., 1999), which measures both discrimination and calibration of predictions.

**Results**  We compare the C-index and IBS of five variants of the Cox PH model: one with an elastic net (EN) penalty, one with our proposed LIT-LVM using a latent distance model, one using PCA to obtain low-dimensional latent embeddings for reconstructing interaction weights after using the elastic net, and one with sparse FMs. We also provide a Random Survival Forest (RSF) model (Ishwaran et al., 2008) as another reference for the accuracy achievable by a nonlinear predictor that is typically one of the best performers among survival prediction algorithms.

Table 3 shows that LIT-LVM outperforms the all other variants of Cox PH models in discrimination and even the much more complex random survival forest (RSF) on the C-index (0.630 vs. 0.629), yet also achieves the best calibration with the lowest integrated Brier score of 0.143. The discordance for RSF, having the second-best C-index but worst IBS (0.156) is not surprising, as the C-index measures ranking only, whereas the Brier score also measures calibration, and tree ensembles such as RSF have been found to suffer from poorer calibration than Cox PH models in other work (Steele

Table 3: Mean C-index and mean Integrated Brier Score (IBS) $\pm$ 1 standard error for survival time prediction over 10 train/test splits. Highest accuracy for each data set is denoted in **bold**, and entries within 1 standard error are denoted in *italic*.

| Model | Mean C-index | Mean IBS |
|---|---|---|
| Cox PH (EN) | $0.627 \pm 0.001$ | $0.149 \pm 0.000$ |
| Cox PH (LIT-LVM) | $\mathbf{0.630 \pm 0.001}$ | $\mathbf{0.143 \pm 0.000}$ |
| Cox PH (PCA) | $0.626 \pm 0.001$ | $0.149 \pm 0.000$ |
| Cox PH (FM) | $0.623 \pm 0.004$ | $0.149 \pm 0.000$ |
| RSF | *$0.629 \pm 0.001$* | $0.156 \pm 0.000$ |

et al., 2018). Notably, the $\approx 0.006$ ($\approx 4\%$) reduction in IBS that LIT-LVM achieves over the elastic net Cox is an order of magnitude larger than the $\leq 0.001$ differences among the other Cox PH variants, underscoring a clear calibration advantage. The difference in IBS between LIT-LVM and the other models is further explained in Section C.4.2 by examining the Brier score over time. LIT-LVM is the only model that combines superior discrimination with superior calibration.

**HLA Latent Representations**  One of the primary objectives of this study is to capture a latent representation of donor and recipient HLAs to model their compatibilities, as greater compatibility is associated with longer survival times. To achieve this, we visualize HLA compatibilities through their distances in the latent space. Initially, we train an elastic net model and used its weights to initialize $\tilde{\beta}$ in LIT-LVM. To initialize the latent positions for different HLA loci, we use the multidimensional scaling approach proposed by Huang & Xu (2022). As a comparative baseline, we construct HLA latent representations using

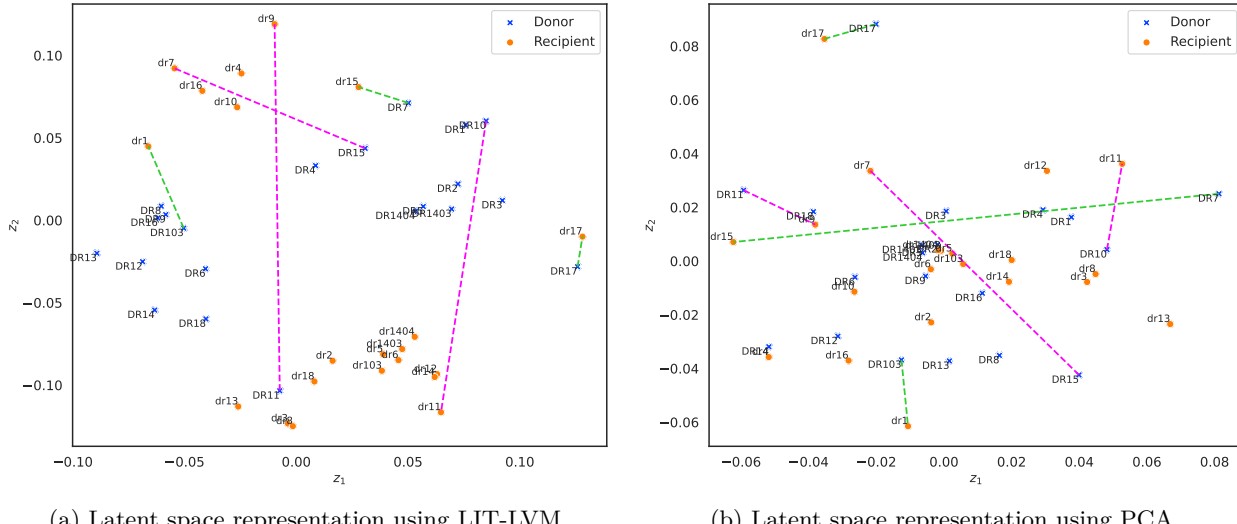

(a) Latent space representation using LIT-LVM
    (b) Latent space representation using PCA

Figure 5: Latent space representation of HLA-DR pairs. Green and magenta dashed lines denote HLA pairs associated with the best (most compatible) and worst (least compatible) outcomes, respectively. In the LIT-LVM latent space, compatible HLA pairs are positioned closely and incompatible pairs are widely separated, whereas PCA fails to preserve this property.

PCA on the weights of HLA pairs learned by Cox PH with elastic net regularization, denoted by Cox PH (PCA) in Table 3.

We plot the latent positions of donor and recipient HLA-DRs in Figure 5 and draw dashed lines between the three HLA pairs with the most negative weights and most positive weights, according to the Cox PH model with only elastic net penalty. In the Cox PH model, negative weights are associated with a lower risk of the occurrence of an event, so they are indicative of higher donor-recipient compatibility. Thus, we expect that HLA pairs with the most negative weights should be positioned closely in the latent space, while those with the most positive weights should be positioned farther apart. As shown in Figure 5a, the top three pairs with the lowest weights ($\theta_{\text{DR17\_dr17}} = -0.068$, $\theta_{\text{DR7\_dr15}} = -0.055$, $\theta_{\text{DR103\_dr1}} = -0.050$) are close together in the latent space, while the top three pairs with the highest weights ($\theta_{\text{DR15\_dr7}} = 0.044$, $\theta_{\text{DR11\_dr9}} = 0.041$, $\theta_{\text{DR10\_dr11}} = 0.040$) are far apart for LIT-LVM. Unlike LIT-LVM, PCA-derived HLA embeddings in Figure 5b fail to place all compatible pairs closely—for instance, DR7_dr15, the second most compatible pair, are widely separated. Furthermore, the PCA embeddings also place many nodes extremely close together near the origin, which does not provide an informative visualization compared to the LIT-LVM embeddings. Latent space representations for HLA-A and HLA-B are shown in Section C.4.2.

Our findings show that our LIT-LVM approach with latent distance model effectively captures the HLA compatibilities that influence kidney graft survival through distances between the latent positions of the HLAs, in addition to achieving prediction accuracy competitive with and superior than more complex ensemble methods such as random survival forest.

# 6 Conclusion

Our underlying hypothesis in this paper was that the coefficients for interaction terms in linear predictors possess an approximate low-dimensional structure. We imposed this approximate low-dimensional structure on the interaction coefficient matrix by representing each feature using a latent vector and penalizing deviations of the estimated interaction coefficients from their expected values modeled by the latent structure. After a thorough investigation on a wide variety of simulated and real datasets, we found that our proposed LIT-LVM approach, which can be viewed as a form of structured regularization specifically for interaction coefficients, generally outperforms elastic net regularization for linear predictors, as well as factorization

machines. The improvement seems to increase with the ratio $p^2/n$; thus, LIT-LVM seems particularly well suited for moderate and high-dimensional settings.

**Limitations**  In this paper, we limited our investigation to second-order interaction terms. Some datasets may exhibit higher-order interactions; our proposed approach could potentially be extended to incorporate these, with the interaction coefficient matrix $\Theta$ replaced by a higher-order tensor. The approximate low-dimensional structure could then be enforced using tensor factorization approaches (Kuleshov et al., 2015). Our LIT-LVM approach also excludes terms of the form $x_j^2$, which can be viewed as self interactions and are usually included in polynomial features. These self interactions could be added into LIT-LVM as terms on the diagonal of $\Theta$ and by choosing a different type of low-dimensional structure that incorporates these terms. Our proposed approach also only scales to thousands of features, as it requires estimation of the explicit $p \times p$ interaction matrix $\Theta$. Problems with even higher values of $p$ are sometimes called ultrahigh-dimensional problems and require methods such as factorization machines or the penalized interaction estimation (PIE) technique proposed by Wang et al. (2021).

### Acknowledgments

Research reported in this publication was supported by the National Library of Medicine of the National Institutes of Health under Award Number R01LM013311 as part of the NSF/NLM Generalizable Data Science Methods for Biomedical Research Program. The content is solely the responsibility of the authors and does not necessarily represent the official views of the National Institutes of Health. This material is also based upon work supported by the National Science Foundation grant IIS-2318751.

The data reported here have been supplied by the Hennepin Healthcare Research Institute (HHRI) as the contractor for the Scientific Registry of Transplant Recipients (SRTR). The interpretation and reporting of these data are the responsibility of the author(s) and in no way should be seen as an official policy of or interpretation by the SRTR or the U.S. Government.

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

# A  Additional Model Information

## A.1  Targeted Interactions

Inclusion of the interaction terms between all the features might not be well suited given the conditions and the domain pertinent to a specific application. Instead, we may be interested in incorporating the interaction between a first set of features $\mathbf{X}' \subseteq \mathbf{X}$ with dimension of $r$ and a second set of features $\mathbf{X}'' \subseteq \mathbf{X}$ with dimension of $q$. We refer to these as *targeted interactions*, which may come from domain knowledge or specific interest in a set of interactions. With targeted interactions, the interaction matrix $\mathbf{\Theta}$ changes from a square $p \times p$ to a possibly rectangular $r \times q$ matrix representing the coefficients for the interaction terms between the features of $\mathbf{X}', \mathbf{X}''$. We denote the latent representation matrices of the two sets of features $\mathbf{X}', \mathbf{X}''$ as $\mathbf{Z}'_{r \times d}, \mathbf{Z}''_{q \times d}$, respectively.

To account for targeted interactions, it is crucial to incorporate a method that selectively prevents updates to the model components representing these non-existent interactions. This selective update is implemented using a mask matrix $\mathbf{M}$, which comprises elements valued at 0 or 1. The value of 0 is assigned to positions

corresponding to non-interacting feature pairs, and 1 is assigned elsewhere. The use of this mask effectively ensures that updates during the training process are only applied to the relevant elements of the interaction weights and their corresponding latent representations.

Implementing a mask matrix provides a controlled mechanism for training a model while preserving the integrity of the domain knowledge within the model's structure. It helps in maintaining the sparsity of the interaction matrix $\boldsymbol{\Theta}$ and the latent representation matrices $\mathbf{Z}'$ and $\mathbf{Z}''$, thus preventing the model from fitting to spurious interactions that are not of interest. This is particularly crucial when the dimensionality of the feature space is high, where there is a risk of overfitting on spurious interactions. Accordingly, the loss functions for the low-rank and latent distance models are modified to incorporate the mask matrix, ensuring that only the targeted interactions contribute to the loss calculation.

### A.2 Model Complexity

Our LIT-LVM model is parameterized by the $(p+1)$-dimensional coefficient vector $\boldsymbol{\beta}$, the $p \times p$ interaction matrix $\boldsymbol{\Theta}$, and the $p \times d$ matrix of latent positions $\mathbf{Z}$. Since the latent dimension $d < p$, there are a total of $O(p^2)$ parameters, which is the same number of parameters that a usual linear predictor with interaction terms would have. On the other hand, FMs store only a factorized representation of $\boldsymbol{\Theta}$ using $\mathbf{Z}\mathbf{Z}^T$, so they have only a total of $O(pd)$ parameters, resulting in a smaller trained model.

The time complexity of our estimation procedure for LIT-LVM per epoch is $O(np^2 + p^2 d) = O(np^2)$ as $np^2$ dominates $p^2 d$. This is the same $O(np^2)$ time complexity as a usual linear predictor with interaction terms, so there is also no asymptotic increase in computation. FMs can again achieve a lower $O(npd)$ time complexity due to their lower number of model parameters.

## B Additional Details and Results on Simulation Experiments

### B.1 Linear Regression with Low Rank Model

We repeat the experiment from Section 4.1 for true latent dimensions of 5 and 10, with results shown in Figure 6. The same trends as in Figure 2a can be observed, with the phase transition around $p = 50$.

Next, we compare our results for different values of $d$, with results shown in Figure 7. For each value of $d_{\text{true}}$, the lowest RMSE is achieved for the correctly specified $d = d_{\text{true}}$, as one would expect. When $d_{\text{true}} = 5$, choosing $d = 2$, which is too small, takes away most of the gain achieved compared to just using elastic net, while choosing $d = 10$, which is too high, still results in an tremendous improvement compared to elastic net, although less than for $d = d_{\text{true}} = 5$. This suggests that, when in doubt, it is better to choose a higher rather than lower dimensional latent representation.

### B.2 Logistic Regression with Latent Distance Model

The results when we repeat the experiment from Section 4.2 across different values of $d_{\text{true}}$ are shown in Figure 8. Again, we see superior prediction accuracy for LIT-LVM, although the improvement is lesser at higher values of $d_{\text{true}}$.

Next, we compare our results for different values of $d$, with results shown in Figure 9. For each value of $d_{\text{true}}$, the highest AUC is achieved for the correctly specified $d = d_{\text{true}}$, as one would expect. When $d_{\text{true}} = 5$, There is still an improvement compared to elastic net even with the misspecified $d = 2$ and $d = 10$. Again, choosing $d$ too high rather than too low results in higher accuracy. The same trends can be observed for $d_{\text{true}} = 2$ and $d_{\text{true}} = 10$.

Finally, we explore what happens when we let $p$ grow to an extremely high value so that $\binom{p}{2} \gg n$. The results up to $p = 500$, for which $\binom{p}{2} = 124{,}750$ with $n = 1{,}000$ are shown in Figure 10. Notice that LIT-LVM continues to outperform elastic net until the point where all of the methods are barely better than a random guess at $p = 500$.

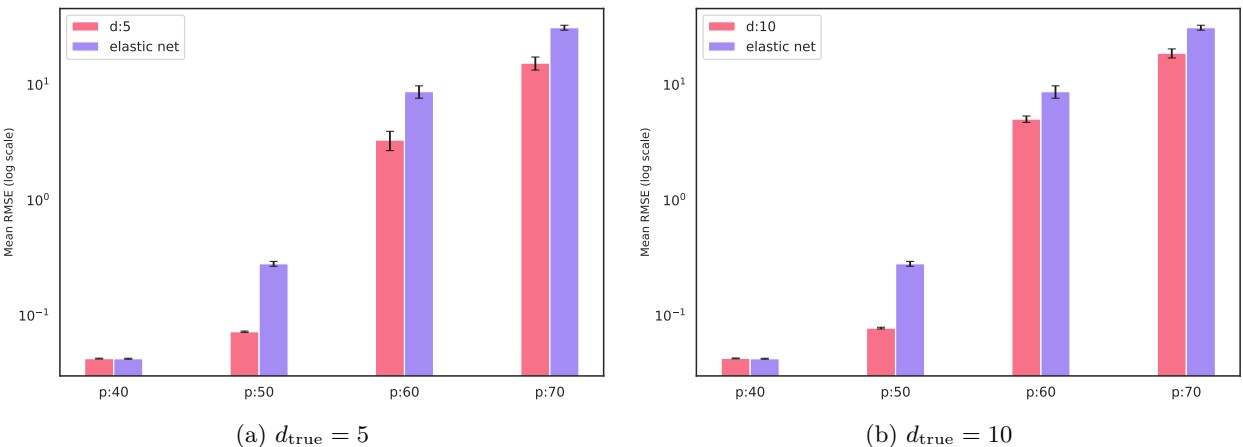

Figure 6: Results of simulation experiments on linear regression for different values of the true latent dimension $d_{\text{true}}$. Lower RMSE denotes higher prediction accuracy. Results for $d_{\text{true}} = 2$ are shown in Figure 2a. Error bars denote one standard error.

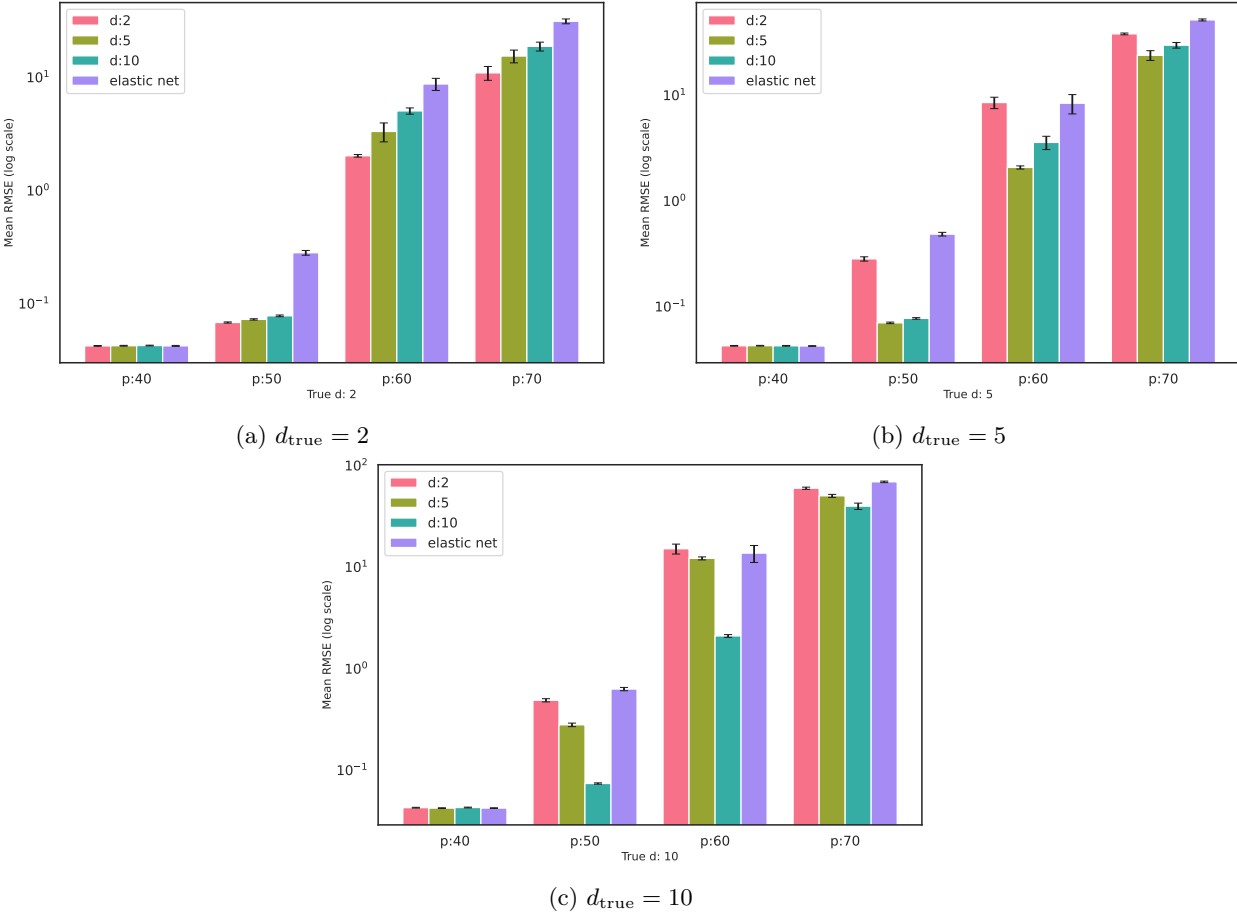

Figure 7: Results of simulation experiments on linear regression for different values of the chosen latent dimension $d$ and true latent dimension $d_{\text{true}}$. Lower RMSE denotes higher prediction accuracy. Error bars denote one standard error.

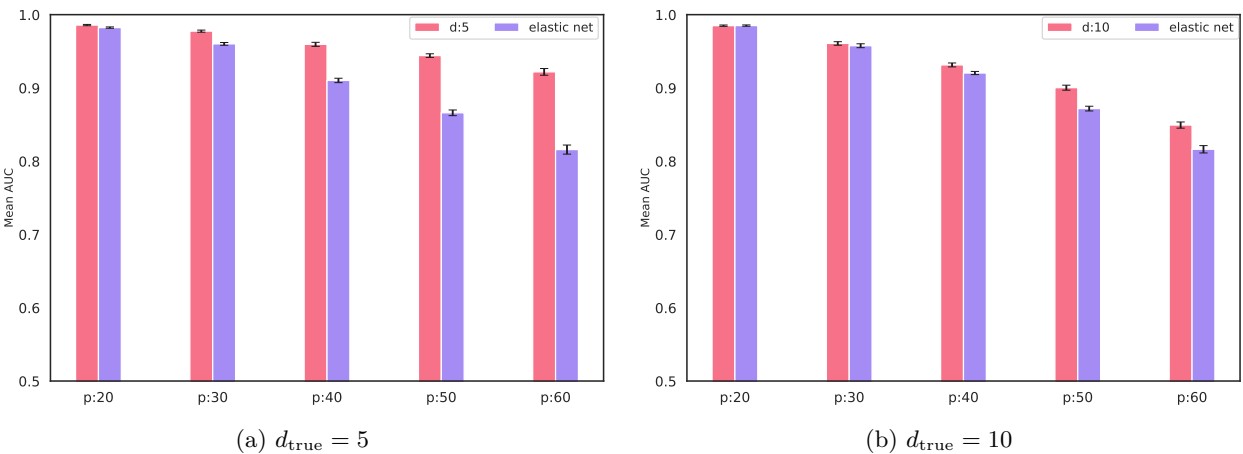

Figure 8: Results of simulation experiments on logistic regression for different values of the true latent dimension $d_{\text{true}}$. Higher AUC denotes higher prediction accuracy. Results for $d_{\text{true}} = 2$ are shown in Figure 2b. Error bars denote one standard error.

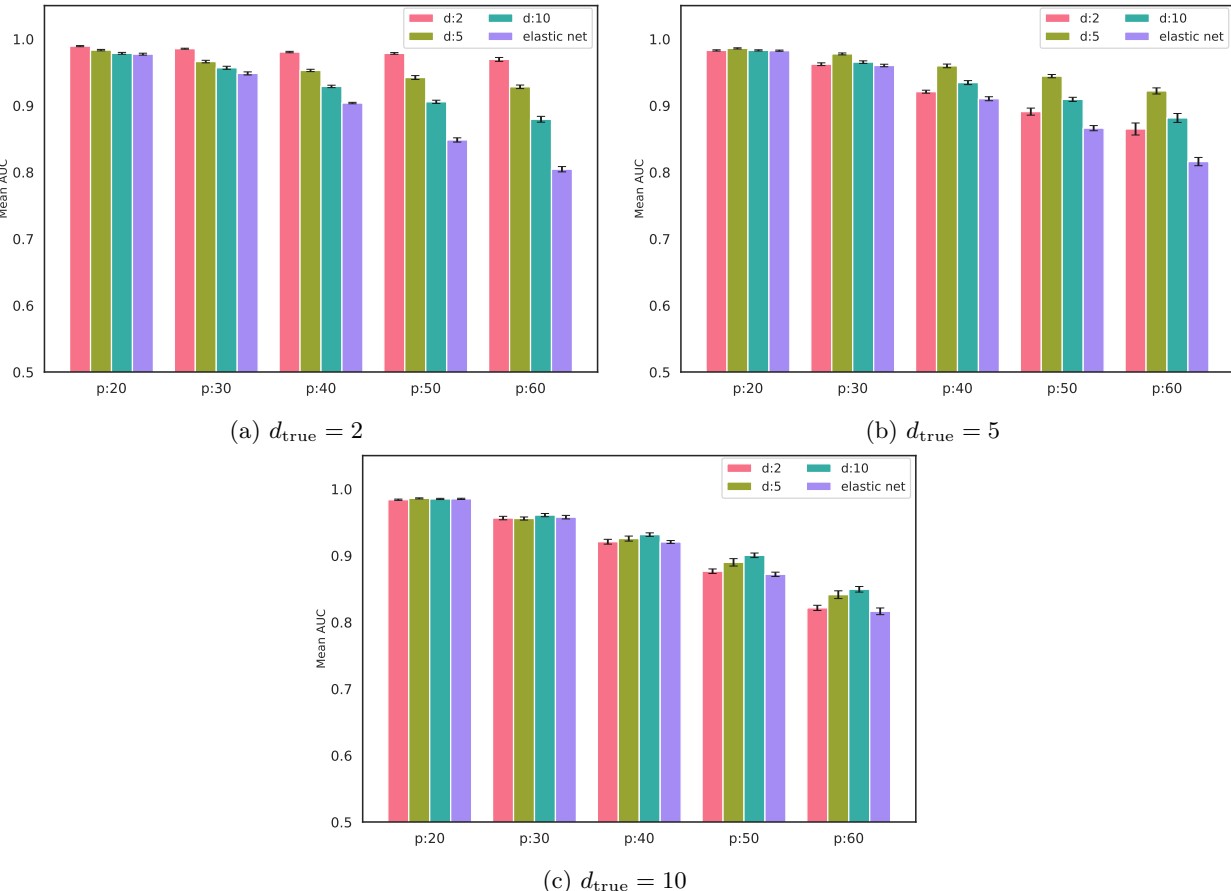

Figure 9: Results of simulation experiments on logistic regression for different values of the chosen latent dimension $d$ and true latent dimension $d_{\text{true}}$. Higher AUC denotes higher prediction accuracy. Error bars denote one standard error.

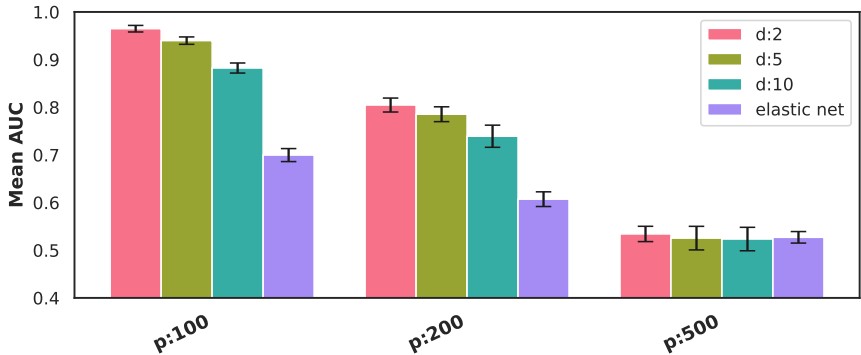

Figure 10: Performance comparison of LIT-LVM vs. elastic net on logistic regression simulation in an extremely high dimension setting. Error bars denote one standard error. LIT-LVM consistently outperforms elastic net up to $p = 500$.

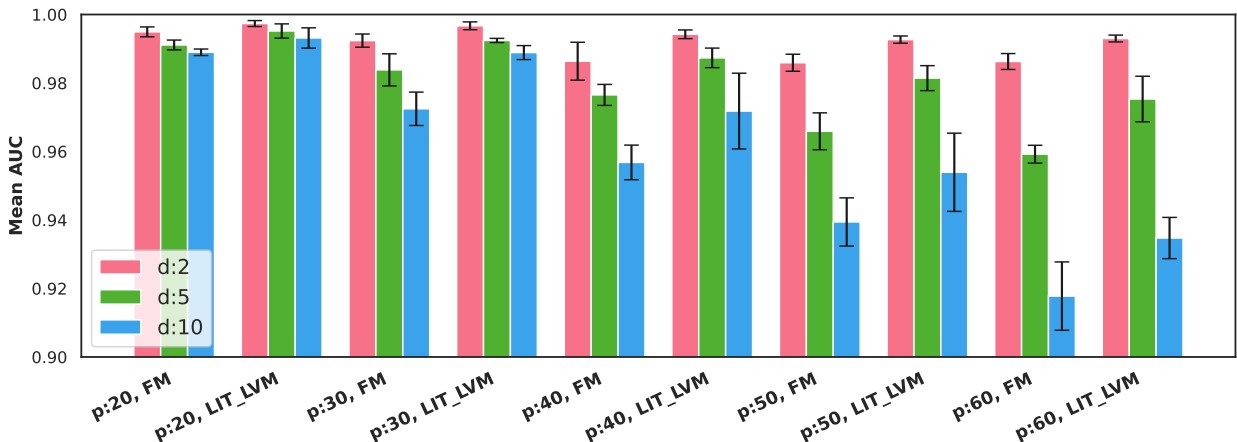

Figure 11: Mean AUC comparison between LIT-LVM and FMs in classification simulation experiments. Error bars denote one standard error. The feature matrix and weights are generated as described in Section 4.2.

### B.3  LIT-LVM vs. Sparse Factorization Machines (FMs)

In this simulation study, we evaluate the performance of sparse FMs compared to LIT-LVM through a series of classification experiments under varied conditions such as increased noise and sparsity injected in the interaction matrix $\mathbf{\Theta}$. These conditions are critical for assessing the performance of the models, as they can potentially affect the predictive accuracy of models like FMs that rely on exact low-rank structures. By introducing greater noise variance and higher sparsity levels in the interaction weights during the simulations, we aim to demonstrate how each model handles deviations from the exact low-rank assumption, thereby providing insights into their practical use in scenarios where this assumption does not hold. Since we are interested in the accuracy of the FM and not the computation time, we formulate FMs as a special case of LIT-LVM with extremely large $\lambda_l = 100,000$. (The model recovered by LIT-LVM is equivalent to that of an FM in the limit $\lambda_l \to \infty$ as discussed in Section 3.3.)

Figure 11 displays the mean AUC results for both models under baseline noise and sparsity conditions, while Figures 12 and 13 show the results when the models are subjected to increased noise levels and greater sparsity. Figure 12 shows the effect of increasing the noise variance to 4 on the performance of the models, highlighting a significant drop in the accuracy of FMs due to their reliance on an exact low-rank structure of the interaction weight matrix. Similarly, Figure 13 demonstrates how increasing the sparsity level in the interaction weights impacts the performance of FMs compared to LIT-LVM. These experiments underscore

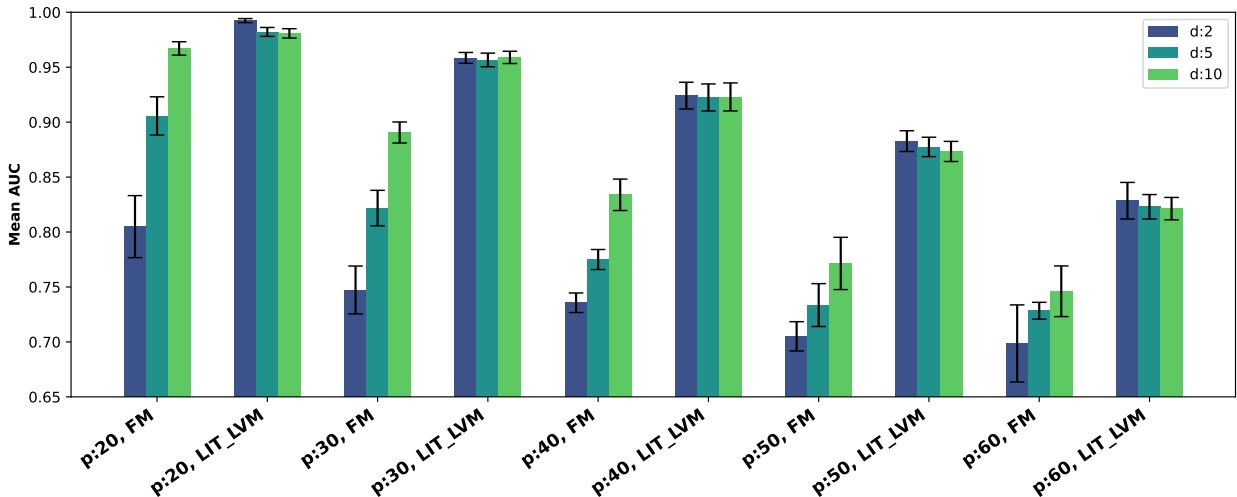

Figure 12: Mean AUC comparison between LIT-LVM and FMs in classification simulation experiments. Error bars denote one standard error. The feature matrix is generated as described in Section 4.2, with increased noise variance (set to 4) added to the true interaction coefficients $\Theta$. Due to this added noise, the performance of FMs, which assume an exact low-rank structure for the interaction coefficient matrix, drops significantly compared to their performance in Figure 11, where the noise was lower.

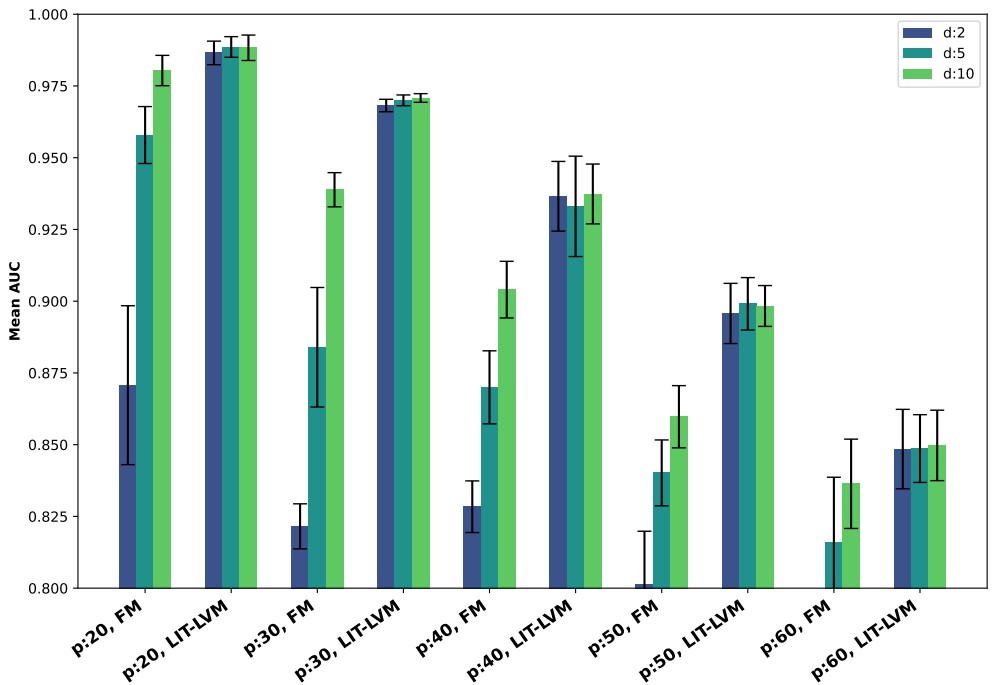

Figure 13: Mean AUC comparison between LIT-LVM and FMs in classification simulation experiments. Error bars denote one standard error. The feature matrix is generated as described in Section 4.2, with increased sparsity level imposed on interaction coefficients $\Theta$. Due to this added sparsity, the performance of FMs, which assume an exact low-rank structure for the interaction coefficient matrix, drops significantly compared to their performance in Figure 11, where the sparsity level was lower.

Table 4: Regression datasets from OpenML used for real data experiments, comprising a wide range of number of samples $n$ and features $p$. Datasets with a slash in the number of features $p$ have categorical variables, so the entry denotes before/after one-hot encoding. The ratio $p^2/n$ is computed using the number of features after one-hot encoding. The OpenML dataset ID column includes a link to the dataset.

| Dataset Name | # of Samples $n$ | # of Features $p$ | $p^2/n$ | OpenML Dataset ID |
|---|---|---|---|---|
| boston | 506 | 14/22 | 0.957 | 531 |
| bike_sharing_demand | 1,737 | 6 | 0.0207 | 44142 |
| diamonds | 53,940 | 10 | 0.00185 | 44140 |
| fri_c4_500_10 | 500 | 10 | 0.200 | 604 |
| medical_charges | 163,065 | 12 | 0.000883 | 44146 |
| nyc-taxi-green-dec-2016 | 581,835 | 9 | 0.000139 | 44143 |
| pol | 15,000 | 26 | 0.0451 | 44134 |
| rmftsa_ladata | 508 | 10 | 0.196 | 666 |
| socmob | 1,156 | 6/39 | 1.32 | 541 |
| space_ga | 3,106 | 7 | 0.158 | 507 |
| superconduct | 21,263 | 79 | 0.294 | 44148 |
| wind | 6,574 | 15 | 0.0342 | 503 |

Table 5: Classification datasets from OpenML used for real data experiments, comprising a wide range of number of samples $n$ and features $p$. The OpenML dataset ID column includes a link to the dataset.

| Dataset Name | # of Samples $n$ | # of Features $p$ | $p^2/n$ | OpenML Dataset ID |
|---|---|---|---|---|
| Bioresponse | 3,434 | 420 | 51.4 | 45019 |
| clean1 | 476 | 169 | 60.0 | 40665 |
| clean2 | 6598 | 169 | 4.33 | 40666 |
| eye_movements | 7,698 | 24 | 0.0748 | 44157 |
| fri_c4_500_100 | 500 | 101 | 20.4 | 742 |
| fri_c4_1000_100 | 1,000 | 101 | 10.2 | 718 |
| hill-valley | 1,212 | 101 | 8.41 | 1479 |
| jannis | 57,580 | 55 | 0.0525 | 43977 |
| jasmine | 2,984 | 145 | 7.05 | 45057 |
| tecator | 240 | 125 | 65.1 | 851 |

the robustness of LIT-LVM under more challenging conditions where assumptions of the exact low-rank structures in FMs do not hold.

# C  Additional Details and Results on Real Data Experiments

## C.1  OpenML Datasets

In our experiments, we utilize a diverse set of benchmark datasets to evaluate the performance of the proposed model. These datasets are chosen to ensure comprehensive coverage of different types of data distributions and characteristics. By using these datasets, we aim to thoroughly validate the performance and generalization ability of our model across both classification and regression tasks.

### C.1.1  Data Description

We employ a range of datasets as detailed in Table 4 for regression and Table 5 for classification. These datasets are selected to test the model's ability to handle different scenarios. The diversity in the number of samples and features ensures that our model is evaluated on a wide spectrum of regression and classification challenges. Links to these datasets are also provided for ease of access.

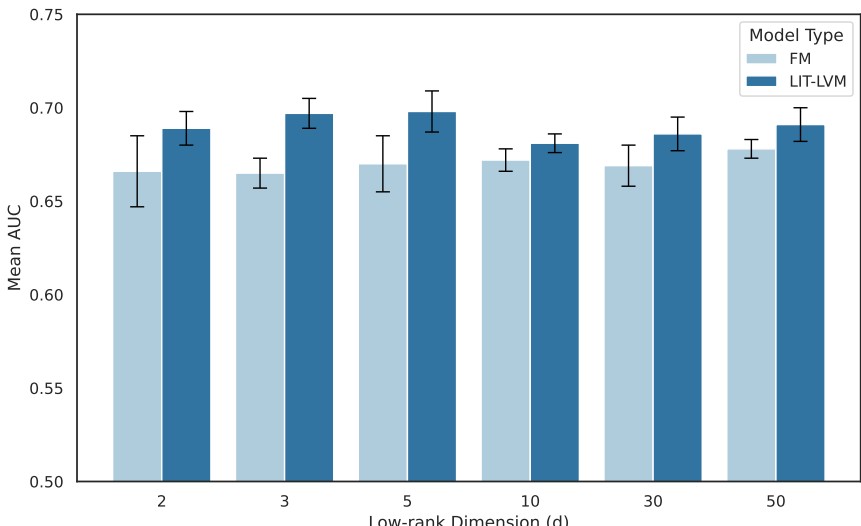

Figure 14: Performance comparison of LIT-LVM vs. FM on `fri_c4_1000_100` classification dataset with different values for the dimension $d$ of the latent representation. LIT-LVM outperforms FM for all values of $d$.

### C.1.2 Hyperparameter Tuning

We ensure that our proposed LIT-LVM approach, elastic net, and FMs are initialized with identical weights, providing a fair comparison. The regularization parameter $\lambda_l$ for the structured regularization in LIT-LVM is chosen from $[0.01, 0.1, 1, 10, 100]$, and the elastic net parameters $\lambda_1$ and $\lambda_2$ are chosen from $[0, 0.01, 0.1, 1, 10, 100]$. For LIT-LVM, we choose fixed dimension $d = 2$ unless otherwise specified. For FMs, we choose $d$ using a grid search over $[2, 10, 25, 50]$, as its assumption of exact rather than approximate low rank structure may require higher $d$.

We use the `hierNet` R package implementing hierarchical lasso (Bien et al., 2013). We tune its regularization parameter $\lambda$ over the same grid $[0.01, 0.1, 1, 10, 100]$ as for the other methods while leaving the $\ell_2$ penalty parameter fixed at $10^{-8}\lambda$, as the authors recommend not to tune this parameter. To match the other methods, we consider only interaction terms and exclude the "self-interactions" (entries on the diagonal of the interaction coefficient matrix).

### C.1.3 Comparison of LIT-LVM with FMs

The comparison between FM and LIT-LVM models in Figure 14 shows that LIT-LVM consistently outperforms FM across all values of the dimension $d$ of the latent representation, with largest performance gap observed at $d = 3$ and $d = 5$. This suggests that LIT-LVM's structured regularization is more effective in estimating interactions, particularly in lower-dimensional latent spaces. FM demonstrates gradual improvement with increasing dimensionality, achieving its best performance at $d = 50$. This suggests that FMs can better approximate the full interaction matrix when the $d$ is larger. This is not surprising because FMs assume an exact factorization of the interaction matrix $\mathbf{\Theta}$, which is likely not exactly low rank in real data sets such as `fri_c4_1000_100`. On the other hand, LIT-LVM's assumption of approximate low rank allows it to better represent the interaction matrix $\mathbf{\Theta}$ using lower dimensions such as $d = 3$ or $d = 5$.

### C.2 Convergence of Optimization Procedure

To complement our empirical evaluations, we analyze the training dynamics of our model across various datasets. Specifically, we plot the training and validation loss curves over epochs in Figure 15, showing consistent and stable convergence of our optimization procedure.

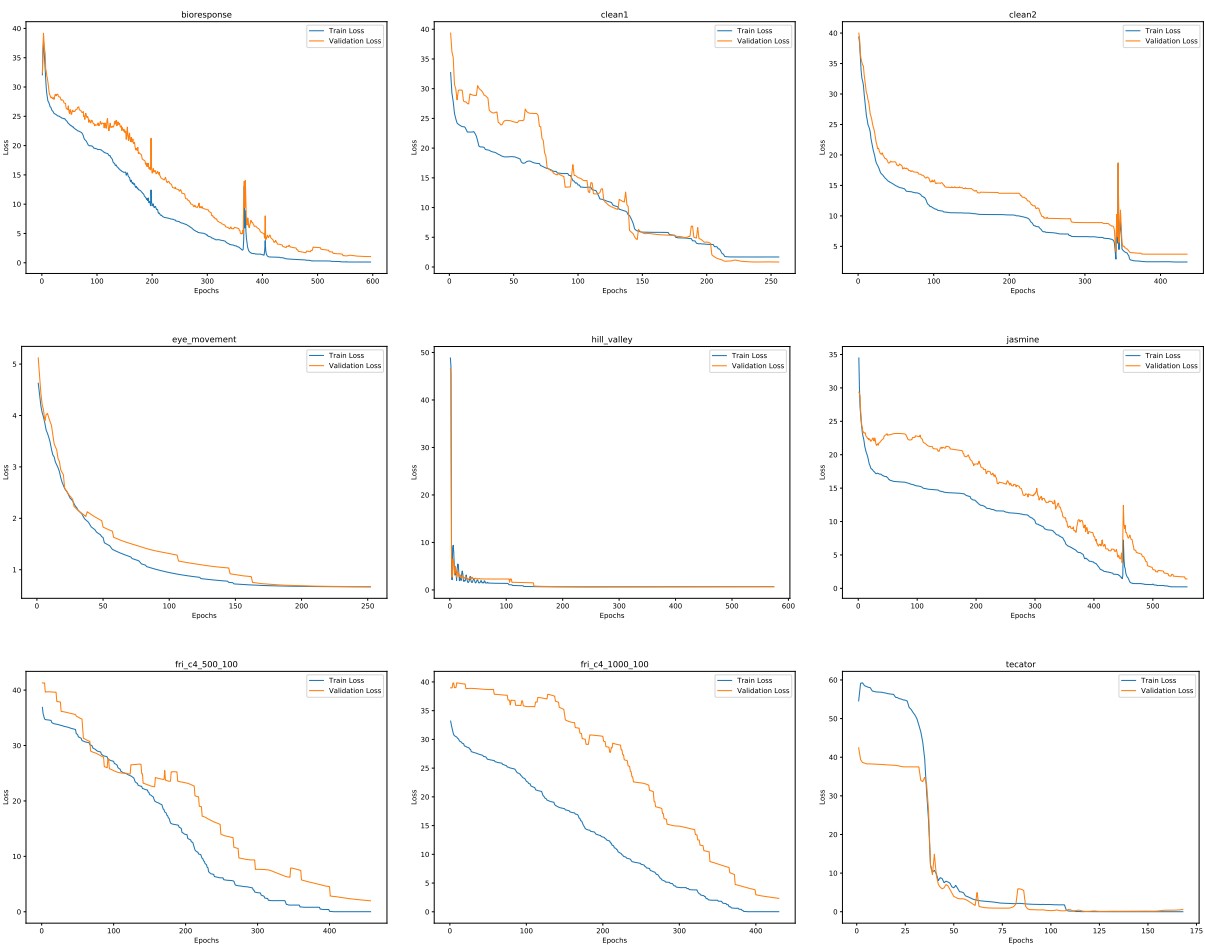

Figure 15: Training and validation loss curves over epochs across different OpenML classification datasets. Each plot shows steady convergence, providing empirical evidence of stable optimization.

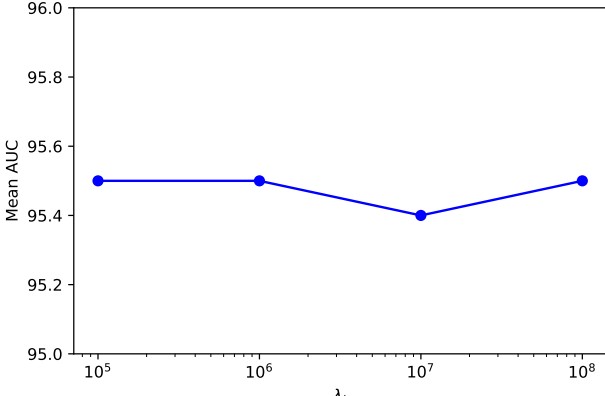

Figure 16: Prediction AUC on the `clean2` dataset as $\lambda_l$ increases beyond $10^5$. The curve is nearly flat, indicating that LIT-LVM behaves like a sparse FM at large $\lambda_l$.

### C.3 Fitting Sparse Factorization Machine Models using LIT-LVM

To further validate that LIT-LVM with large $\lambda_l$ approximates the behavior of a sparse factorization machine (FM), we conduct an additional experiment on the `clean2` dataset. Specifically, we increase $\lambda_l$ values beyond the $10^5$ value that we use in earlier experiments and measure its effect on prediction accuracy. As shown in Figure 16, the AUC remains nearly constant beyond this threshold, confirming that the LIT-LVM model with $\lambda_l = 10^5$ is sufficiently large to represent a sparse FM model.

### C.4 Survival Prediction and Compatibility Estimation for Kidney Transplantation

#### C.4.1 Data Preprocessing

For this study, we represent each HLA as a categorical feature which we call an HLA type. We define *HLA pairs* to represent the interactions between donor and recipient HLA types. The HLA type and pair features are encoded using the approach of Nemati et al. (2023), which we summarize in the following.

HLA types are encoded using a one-hot-like scheme, producing binary variables such as `DON_A1`, `DON_A2`, `REC_A1`, and `REC_A2`. The binary variable is set to one if the donor or recipient possesses the corresponding HLA type. We refer to the encoding as one-hot-like because some HLA types are splits of a broad HLA type. For example, A23 is a split of A9, so that each occurrence of A23 in a donor is encoded with a one in both columns `DON_A23` and `DON_A9`.

HLA pairs are similarly represented with binary one-hot-like variables such as `DON_A1_REC_A1`, `DON_A1_REC_A2`, etc., again accounting for broads and splits. A value of one in a particular column indicates that both the donor and recipient possess the specified HLA types, and the donor-recipient HLA pair is biologically relevant. Some HLA pairs are not biologically relevant due to the asymmetry between donors and recipients—if a donor possesses an HLA type that is not in the recipient, then the recipient's immune system may reject the transplant, so the HLA pair is biologically relevant for evaluating donor-recipient compatibility. On the other hand, if a recipient possesses an HLA type that is not present in the donor, it does not create a problem for the recipient, so the HLA pair is not biologically relevant, and we do not place a one in the column for that HLA pair. We refer interested readers to Nemati et al. (2023) for further details and illustrative examples of the encodings.

#### C.4.2 Additional Results

A plot of Brier score over time is shown in Figure 17. Within a short period of time just under 1 year, RSF already shows worse Brier scores than the other approaches and has worse calibration over the entire time horizon. The Cox PH variants except for LIT-LVM all have roughly equal Brier scores. Beyond about 5 years, LIT-LVM shows noticeable improvement compared to the other Cox PH models, and this improvement persists over the entire time horizon, resulting in the lower integrated Brier score for LIT-LVM shown in Table 3.

**HLA Latent Representations** The latent representations for HLA-A and HLA-B are shown in Figure 18. We make similar observations to those for HLA-DR in Figure 5. First, notice that the PCA embeddings for HLA-A and HLA-B tend to place almost all of the HLAs together around the origin, even more so than for HLA-DR. This decreases the interpretability of the embedding, especially compared to the LIT-LVM embedding, where the HLAs are more spaced apart. Furthermore, the distances between low compatibility and high compatibility are also not well preserved in the PCA embedding compared to the LIT-LVM embedding. For HLA-A, the most compatible pairs are ($\theta_{\text{A68\_a2}} = -0.053$, $\theta_{\text{A2\_a24}} = -0.052$, $\theta_{\text{A2\_a68}} = -0.050$), which are placed in the vicinity of each other, while the least compatible pairs are ($\theta_{\text{A32\_a74}} = 0.050$, $\theta_{\text{A1\_a3}} = 0.044$, $\theta_{\text{A2\_a3}} = 0.040$), which are placed far from each other. Similarly, for HLA-B, the most compatible pairs are ($\theta_{\text{B8\_b60}} = -0.010$, $\theta_{\text{B58\_b8}} = -0.006$, $\theta_{\text{B44\_b35}} = -0.005$), which are placed in the vicinity of each other, while the least compatible pairs are ($\theta_{\text{B60\_b35}} = 0.007$, $\theta_{\text{B35\_b39}} = 0.007$, $\theta_{\text{B53\_b72}} = 0.005$), which are placed far from each other.

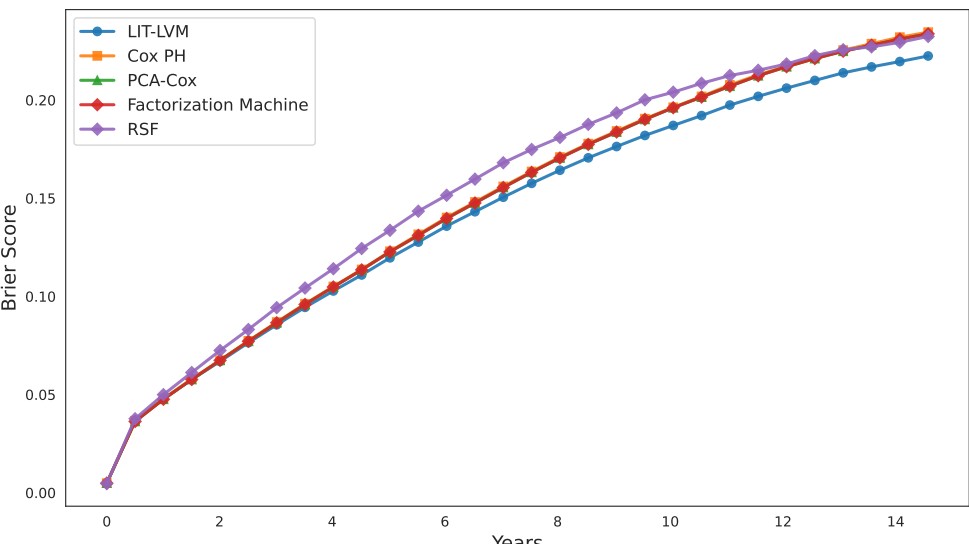

Figure 17: Comparison of Brier scores over 30 time points. LIT-LVM achieves the best calibration, as indicated by the lowest Brier score. The improvement is greatest at high survival times, beyond the median survival time of about 10 years.

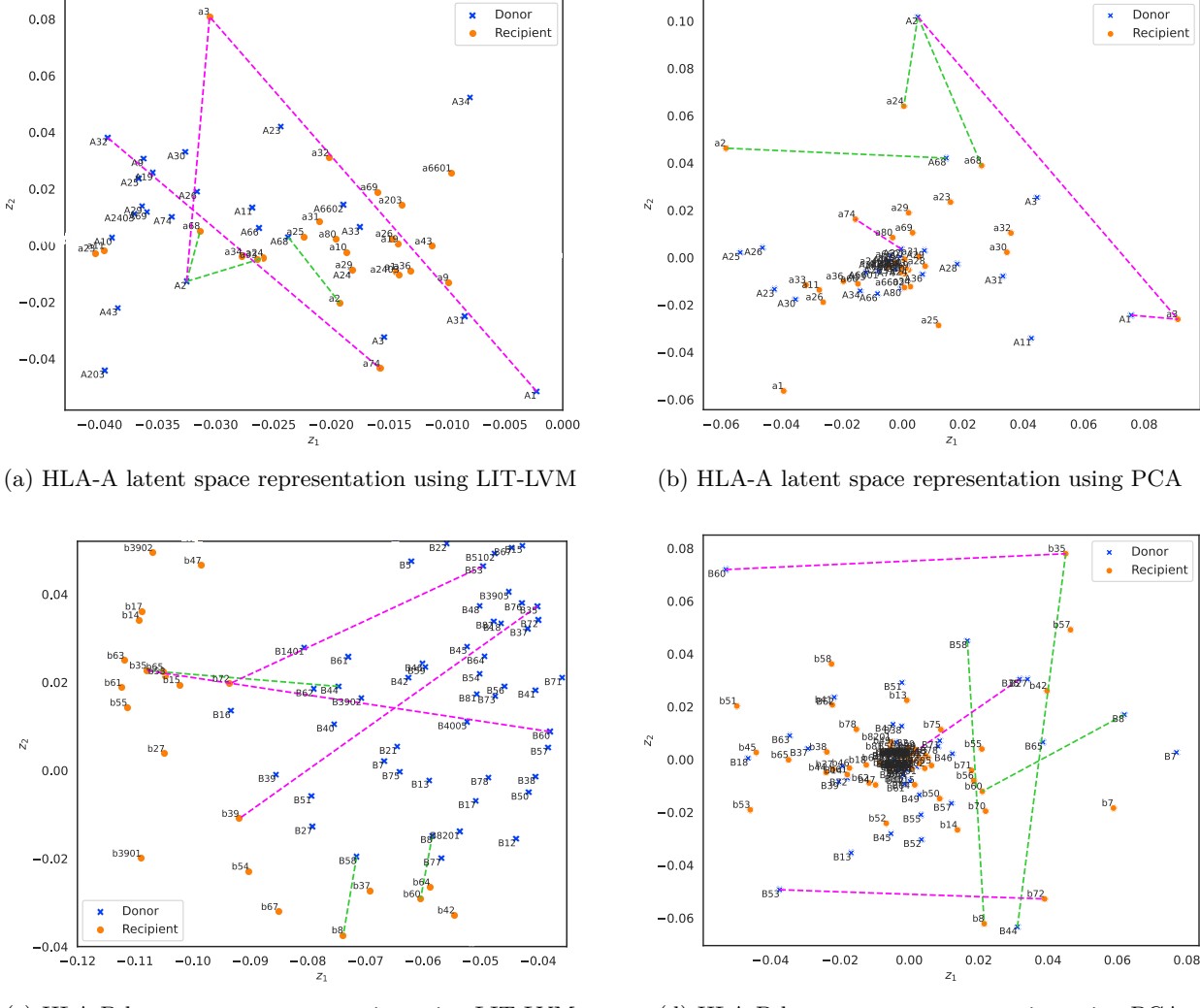

(a) HLA-A latent space representation using LIT-LVM

(b) HLA-A latent space representation using PCA

(c) HLA-B latent space representation using LIT-LVM

(d) HLA-B latent space representation using PCA

Figure 18: HLA-A and HLA-B latent positions from LIT-LVM and PCA, highlighting the top three most compatible and least compatible HLA pairs according to the elastic net penalized Cox PH model's coefficients. Green dashed lines denote the most compatible pairs, and magenta dashed lines denote least compatible pairs. In LIT-LVM, the most and least compatible pairs are placed close together and far apart, respectively, while the PCA latent spaces fail to preserve this property.

