# OpenReview forum: "LIT-LVM: Structured Regularization for Interaction Terms in Linear Predictors using Latent Variable Models"
_TMLR — Accepted by TMLR_

### Review · Reviewer_QNQs · 2025-06-18

**Summary Of Contributions:**

The authors look at linear models enhanced with pair-wise interaction features (product terms like x_i*x_j).
Those interactions quickly explode in the number of features p give p(p-1)/2 extra coefficients—so even strong sparsity penalties such as the lasso or elastic-net start to overfit when p^2\geq n.

Their key observation is that the matrix of interaction weights x_i*x_j for  (which is a p*p upper-triangular matrix), is rarely arbitrary: many of the pair-wise effects are correlated because the underlying features share common “roles” or “types”. Formally, they hypothesise an approximate low-dimensional structure in the matrix and capture it with a latent-variable model (LVM)

The main approach is to assign every feature a latent (but lower dimensional d<p) vector z_j, and then the targeted model requires each interaction weights to sit near a simple function of  a dot-product  or  pair-wise distance.

**Audience:**

Yes

**Broader Impact Concerns:**

None.

**Claims And Evidence:**

Yes

**Requested Changes:**

I think the main issue for the current paper is its lack of discussion to related researches, to name a few:

Projection-pursuit chases non-linear ridge functions of projections; LIT-LVM stays linear but says: “if you must keep all x_i*x_j terms, at least tie their coefficients together through a low-dimensional latent map so they don’t overfit.” I am not sure I understand this, how is the model fundamentally different from projection pursuit, or it is a revised version with sparsity penalty?
c.f. Friedman, Jerome H., and Werner Stuetzle. "Projection pursuit regression." Journal of the American statistical Association 76.376 (1981): 817-823.

Deep-kernel methods use a neural network as an unknown feature extractor before a Gaussian-process kernel; they can model very rich surfaces but give up the transparency of explicit \beta and \Theta. LIT-LVM keeps every coefficient on the table and simply nudges them toward a coherent latent pattern, are these two methods related or .
c.f., Noack, Marcus M., Hengrui Luo, and Mark D. Risser. "A unifying perspective on non-stationary kernels for deeper Gaussian processes." APL Machine Learning 2.1 (2024).
Wilson, Andrew Gordon, et al. "Deep kernel learning." Artificial intelligence and statistics. PMLR, 2016.

I think that the proposed model fills a niche between classical linear-interaction models (too many parameters), low-rank factorization (too rigid), and fully non-parametric surfaces (hard to interpret), but this is not clear in the current draft; specifically if the authors do not provide theoretic guarantees (e.g., consistency, risk bounds), then the experimental evidence for comparing its performance against related methods above, and state-of-the-art method is insufficient in the current shape.

(1)Requested changes: can the authors add discussion to related literature and methods, then append experimental evidence to illustrate their method?

(2)Requested changes 2: One of the benefit of LIT model is illustrated in "HLA Latent Representations" application, it only compares its outcome with PCA; Admittedly, it is not too popular to perform dimension reduction on latent variable space, yet there are certain line of research called supervised dimension reduction (e.g., SIR), are these relevant and worth compared to? In addition, can we use other unsupervised dimension reduction methods to have a more comprehensive comparison?

**Strengths And Weaknesses:**

Strengths

(1)The proposed model directly tackles the p^2>n problem in linear interaction model by softly shrinking the interaction matrix toward a low-rank or latent-distance form rather than enforcing an exact factorisation.

(2)It works with linear, logistic and survival objectives and uses an Adam-plus-proximal optimiser that avoids any SVDs while keeping per-epoch cost at O(n*p^2), which does not seem to be too expensive for big p.

(3)The learned latent vectors create interpretable maps (e.g., HLA compatibility clusters), and the authors provide code and hyper-parameter grids to aid reproducibility.

Weaknesses
(1) Model quality hinges on four hyper-parameters (\lambda_1,\lambda_2,\lambda_\ell,d), yet the paper offers minimal tuning guidance.

(2) If the true interaction matrix is near full-rank (like network-type covariate), the latent penalty may bias estimates, and robustness is demonstrated only on synthetic data. I do not see how to avoid this. In this scenario, the framework handles pairwise interactions only, leaving higher-order effects unexplored (which is pretty common in ANOVA type models).

(3) See requested changes, for the detailed explanation for the weakness in experimental part.

---

> ### Author Response · Authors · 2025-08-02
> **Response to Reviewer QNQs**
>
> We thank the reviewer for the constructive feedback. We address each point below and have revised the paper accordingly.
>
> # 1. Hyperparameter Tuning Guidance
>
> **Response**: We suggest fixing $d=2$ across datasets, as our experiments (Tables 1–2) show that low-dimensional embeddings already achieve strong performance, often making $d$ tuning unnecessary. For $\lambda_l$, we recommend a small grid such as ${0.01, 0.1, 1, 10}$; our sensitivity analysis in Section 4.2 (Figures 3–4) shows that moderate values optimize AUC when latent structure exists, while smaller values perform better when it does not. $\lambda_1$ and $\lambda_2$ can be tuned using standard log-scale cross-validation grids (e.g., powers of 10), as is common for the elastic net. With this guidance, fixing $d$, limiting the $\lambda_l$ grid, and applying standard search for $\lambda_1$ and $\lambda_2$ keeps LIT-LVM's tuning effort manageable.
>
>
> # 2. Full-Rank Interaction Matrix Robustness
>
> **Response**: We assume only that $\Theta$ is close to low rank in Frobenius norm, not exactly low rank. It may be high-rank due to the error term $\boldsymbol{\epsilon}$. We agree that the latent penalty increases bias and may degrade accuracy if $\lambda_l$ is too high, which is an intrinsic tradeoff in regularized models. In Figure 3b, we simulate a high-noise setting ($\sigma^2_\theta = 4$), making $\Theta$ effectively full-rank. When $\lambda_l = 0.01$, LIT-LVM matches the accuracy of elastic net. Higher $\lambda_l$ lowers accuracy, but our tuning recommendations help detect and avoid this. Thus, LIT-LVM is a flexible regularizer that aids when structure exists and has minimal effect when it does not.
>
> # 3. Limitation to Pairwise Interactions
>
> **Response**: As mentioned in the paper, we focus on pairwise terms due to the balance between interpretability and scalability (Section 2.2.1). However, the approach is extendable: replacing the matrix $\Theta$ with a tensor $T \in \mathbb{R}^{p \times \dots \times p}$ allows modeling $m$-way interactions, with a factorized penalty like $\\|T - \hat{T}\\|_F^2$.
>
> # 4. Difference from Projection Pursuit Regression
>
> **Response**: PPR models $y = \sum f_j(a_j^T x)$ via ridge functions, which are flexible but not interpretable. LIT-LVM keeps the full linear structure with explicit $\beta$ and $\Theta$. It models individual pairwise effects with latent regularization, preserving coefficient-level interpretability. This is especially useful in biomedical applications.
>
> # 5. Relation to Deep Kernel Learning
>
> **Response**: DKL uses neural nets to learn deep features for Gaussian processes. It captures rich nonlinearities but does not retain interpretable parameters. In contrast, LIT-LVM models pairwise interactions explicitly, constraining them through a learned latent map while keeping all coefficients interpretable. The goals and use cases are different: LIT-LVM targets structured generalization and interpretability.
>
> # 6. Theoretical Guarantees
>
> **Response**: We acknowledge the lack of theoretical guarantees and consider this an important direction for future work. To support the validity of our method, we provide strong empirical results across diverse simulated and real datasets. Additionally, we include training and validation loss curves in Figure 15 to illustrate learning behavior. These plots demonstrate smooth convergence, even though different datasets are trained for varying numbers of epochs based on our stopping criteria. This reflects how the model adapts naturally to each dataset's complexity. The observed consistency across tasks offers practical evidence of robustness, effective generalization, and optimization stability, reinforcing the reliability of our approach even in the absence of formal theory.
>
> # 7. Expand Latent Representation Comparison Beyond PCA
>
>
> **Response**: We expanded the comparison to include multidimensional scaling (MDS), t-SNE, and autoencoders (2D bottleneck). Results (C-index):
>
> - MDS: $0.627 \pm 0.001$
> - Autoencoder: $0.627 \pm 0.001$
> - t-SNE: $0.521 \pm 0.001$
>
> LIT-LVM outperforms all of these methods.
>
> Supervised DR methods are not applicable here, as our embeddings are for the features and not the samples. There is no class label for the features to use as supervision.
>
> # 8. Clarifying LIT-LVM’s Modeling Niche
>
> **Response**: We agree with the reviewer’s assessment that LIT-LVM fills the space between
>
> - Elastic net: sparse but unstructured.
> - Factorization machines: exact low-rank but rigid.
> - Nonparametric models: more expressive but not interpretable.
>
> LIT-LVM provides soft structured regularization, improving performance when $p^2/n$ is large while maintaining interpretability. This is emphasized in Sections 2.2.3 and 5.
>
> We do not claim superior performance compared to the general class of non-linear models. Rather, we only claim superior performance to models that incorporate only linear and pairwise interaction terms, which forms our basis for comparison methods.

---

> > ### Comment · Reviewer_QNQs · 2025-08-02
> >
> > Based on TMLR's criterion of acceptance, I have no further comments, but I hope the author can explain more about interpretability, since this seems like a key design choice to their point 3 and 5.

---

### Review · Reviewer_zEr4 · 2025-07-01

**Summary Of Contributions:**

This paper introduces a novel regularization approach (LIT-LVM) designed to accurately estimate coefficients for interaction terms in linear predictors. LIT-LVM is founded on the hypothesis that the coefficients for different interaction terms possess an approximate low-dimensional structure, and it models this by representing each feature with a latent vector in a low-dimensional space. By incorporating a latent variable model term into its total loss function, which penalizes deviations from this hypothesized latent structure, LIT-LVM achieves superior prediction accuracy compared to traditional elastic net regularization and Factorization Machines (FMs). Furthermore, it provides interpretable low-dimensional latent representations of features, which are valuable for visualizing and analyzing relationships, as demonstrated in applications such as modeling donor-recipient compatibility in kidney transplantation. The method's robustness to noisy or non-exactly low-rank interaction structures, in contrast to the exact low-rank assumption of FMs, contributes to its enhanced performance on real-world datasets.

**Audience:**

Yes

**Claims And Evidence:**

Yes

**Requested Changes:**

Based on TMLR's acceptance criteria, I have an overall favorable opinion of this paper. However, as noted in the Weaknesses section, I also have several concerns that need to be addressed. To ensure eventual acceptance, I would like to ask your views on these concerns, including appropriate revisions to the paper.

[Minor Comments]

- “such that so that” in the first paragraph of Section 2.2.3;
- $\theta_{\theta}^2$ in the caption of Figure 3;
- “0.0.004” at (‘space_ga’, ‘FM’) in Table 1;
- $\mathrm{logistic}(\beta^{\top}x+\theta_{\mathrm{flat}}^{\top}x_{\mathrm{int}}) + \eta)$ in the first paragraph of Section 4.2.

**Strengths And Weaknesses:**

### Strengths

- **Effective Overfitting Mitigation in High-Dimensional Settings**: LIT-LVM is designed to further mitigate overfitting in high-dimensional environments, especially when the number of interaction terms (on the order of $p^2$) is large compared to the number of samples ($n$). This structured regularization is particularly beneficial as the ratio $p^2/n$ increases.
- **Broad Applicability to Various Linear Predictors**: The LIT-LVM framework is applicable to many types of linear predictors, including linear regression, logistic regression, and the Cox Proportional Hazards (Cox PH) model, and can be used in conjunction with standard regularization techniques like elastic net.
- **Greater Modeling Flexibility due to Approximate Structure**: Unlike Factorization Machines (FMs), which assume an exact low-rank structure for interaction coefficients, LIT-LVM hypothesizes an approximate low-dimensional structure, penalizing deviations from it.
- **Superior Prediction Accuracy**: The approach consistently demonstrates superior prediction accuracy compared to traditional methods like elastic net and factorization machines (FMs) across a wide range of simulated and real datasets for both regression and classification tasks. This improvement is notable, especially as feature dimensions increase.
- **Provides Interpretable Low-Dimensional Latent Representations**: LIT-LVM offers a significant advantage by providing low-dimensional latent representations for features. These representations are useful for visualizing and analyzing relationships between features. For example, in the kidney transplantation study, LIT-LVM's latent space effectively positioned compatible HLA pairs closer and incompatible pairs farther apart, offering valuable interpretability that PCA failed to achieve.

### Weaknesses

**[Low-Dimensional Assumption]**

- The assumption that “this matrix has an approximate low-dimensional structure” (Section 1) would benefit from clarification regarding how it corresponds to specific real-world or hypothetical scenarios.

**[Non-Convexity]**

- Methods that involve latent variable models and approximate low-dimensional structures, such as the LIT-LVM approach described, often lead to non-convex optimization problems. Please clarify whether the proposed method falls into this category.
- If it is non-convex, I have concerns about the following point. The paper mentions that the optimization is performed using Adam. While this is one of the standard optimization algorithms for such problems, the manuscript does not discuss the implications of this non-convexity, nor does it provide any theoretical guarantees or experimental analyses regarding the convergence of the optimization procedure. Since It is unclear how stable and optimizable the proposed method is, this is a significant omission for a method involving a non-convex objective.

**[Flexibility vs. Sparse FMs]**

- There is an inconsistent argument about flexibility. Initially, LIT-LVM is framed as preventing overfitting via structured regularization to counteract the high flexibility of interaction terms. However, in Section 3.3 and the comparison with Sparse FM, it is claimed that LIT-LVM outperforms due to its greater modeling flexibility. Please reconcile these two perspectives.
- In addition to the above, the proposed method and its clear competitor, Sparse FMs, are not compared in numerical experiments.
- In Section 5.2, there is a description of a comparison with Sparse FMs, but the label in Table 3 is Cox PH (FM).
- In any case, providing a consistent and comprehensive comparison against Sparse FMs across all experiments would strengthen the persuasiveness of your claims.

**[Robustness]**

- The numerical experiments in Section 4 and Appendix B evaluate the proposed method using a discrete set of latent dimensions (specifically,  $d=2,5,10$). If $d$ is continuously varied over the natural numbers, does the proposed method still identify the optimal dimension and yield the best results
- Can the proposed method maintain high performance compared to the comparative methods even in situations of model misspecification (e.g., when the interaction term is not $x_jx_k$)?

**[Reproducibility]**

- I attempted to reproduce the numerical experiments using the provided GitHub repository, but there were no usage instructions.
- Could you add the necessary information for execution, such as `README.md` and `requirements.txt`? (Personally, I would appreciate it if you could add it with Pipenv.)

---

> ### Author Response · Authors · 2025-08-02
> **Response to Reviewer zEr4**
>
> We thank the reviewer for the thoughtful and detailed comments.
>
> # 1.Low-Dimensional Assumption
> **Comment**: Clarify how approximate low-dimensional structure corresponds to motivating application settings.
>
> **Response**:
> The assumption that $\Theta$ has an approximate low-dimensional structure is motivated by settings where pairwise interaction coefficients can be captured by a few latent factors, extending the factorization machine assumption that the interaction matrix is exactly low rank.
>
> In our kidney transplant application, HLA compatibility is modeled via distances in a learned latent space, with each HLA type represented as a vector in $\mathbb{R}^d$. This setup allows rarely observed HLA pairs (high-variance estimates) to borrow statistical strength from common ones (low-variance estimates). This approach is supported by prior work: latent distance models for binary networks (Hoff et al., 2002), HLA compatibility modeling (Huang & Xu, 2022), and signed networks (Nakis et al., 2023), as also discussed in our response to Reviewer bGaA. These studies show that latent distance or dot-product structures effectively model complex pairwise interactions
>
> We fix $d=2$ in all experiments, based on empirical evidence that low-dimensional embeddings capture compatibility patterns well. As shown in Section 4 and the latent-noise ablation (Figure 3b), LIT-LVM performs strongly when low-rank structure exists and remains robust when it does not, making it well-suited for domains where interactions are approximately, but not exactly, low-rank.
>
> # 2. Non-Convexity of Optimization Problem
> **Comment**: I have concerns about convergence of the optimization procedure given lack of theoretical guarantees or experimental analyses.
>
> **Response**:
> We confirm that the overall loss is non-convex due to the latent structure terms. While we do not provide formal convergence guarantees, our optimization strategy uses Adam with proximal updates for $\ell_1$ terms and automatic differentiation that yields smooth and stable convergence across datasets. Figure 15 in the appendix shows training and validation losses decreasing and plateauing, offering empirical evidence of convergence. We acknowledge that a theoretical analysis remains an important direction for future work.
>
> ## 3.Flexibility vs. Sparse FMs
>
> **Response**:
> We believe there is a misunderstanding. We claim that LIT-LVM prevents overfitting compared to the elastic net with interactions which penalizes each interaction coefficient separately by imposing structured regularization. We also claim that LIT-LVM has greater modeling flexibility than sparse FMs because it penalizes deviations from low-dimensional structure, rather than imposing such structure via factorization. In terms of modeling flexibility: elastic net with interactions > LIT-LVM > sparse FMs; and in terms of overfitting control: sparse FMs > LIT-LVM > elastic net with interactions.
>
> We also apologize for the confusion with respect to the experiment results. The entries labeled as FMs are actually our implementation of sparse FMs by using LIT-LVM with $\lambda_l = 100{,}000$. (Recall that LIT-LVM approaches a sparse FM in the limit as $\lambda_l \rightarrow \infty$.) We use this approach to avoid the more complex optimization proposed by Atarashi et al. (2021), so that any differences in performance are not caused by the optimizer.
>
> To confirm this choice was sufficient, we tested even larger $\lambda_l$ values (beyond $10^5$) and observed nearly flat AUCs, as shown in Figure 16.
>
> ## 4. Robustness
>
> **Response**:
> We thank the reviewer for highlighting these points. In all of our real data experiments, we fix the latent dimension $d=2$ across datasets. This choice is motivated by the observation that low-dimensional embeddings often are enough to capture interaction structure, and our empirical results (Tables 1–2) confirm that $d=2$ already yields strong performance across diverse tasks. We do not attempt to identify the optimal $d$ but instead show in Appendix B (Figures 7,9) that LIT-LVM remains robust under misspecification of $d$.
>
> If the interaction term is not $x_j x_k$, then it would involve higher-order interactions or other forms of non-linearities. We have not investigated the performance of LIT-LVM against other approaches in this setting, although we believe that its soft regularization-based approach would outperform the FM’s hard factorization-based approach, which strictly enforces the low rank structure. Whether it outperforms elastic net with interactions (which models each interaction coefficient separately) would likely depend on the degree of misspecification of the model, and we believe that this is an interesting area for future research.
>
> ## 5. Reproducibility
>
> **Response**:
> We have added a README.md file containing step-by-step instructions for running all experiments
>
> ## 6. Minor Typos
> **Response**:  We have now corrected the errors and highlighted them in blue.

---

> > ### Comment · Reviewer_zEr4 · 2025-08-21
> >
> > Thank you for the opportunity to review the manuscript. I would like to express my sincere gratitude to the authors for their detailed responses and the considerable effort invested in revising the paper.
> >
> > Overall, the authors have made substantial improvements by addressing the concerns raised and providing valuable additional material. As a result, the paper has become significantly stronger.

---

### Review · Reviewer_bGaA · 2025-07-20

**Summary Of Contributions:**

This paper proposes a structural regularization method for simultaneously estimating the main effects and interaction effects in regression. The idea is to assume that the interaction coefficient matrix has an approximate low-rank structure, which can be captured by a low-rank model or a latent distance model. The paper formulated the optimization problem and conducted experiments on simulated and real data to validate the effectiveness of the proposed method.

**Audience:**

Yes

**Claims And Evidence:**

Yes

**Requested Changes:**

See comments above.

Minor issue:

Beginning of Introduction, “Linear predictors such as linear regression and logistic regression …”: predictors are variables, not models; please rephrase.

**Strengths And Weaknesses:**

**Strengths**

The proposed latent space representations are more flexible than those considered in the literature, and seem to perform well on a wide variety of datasets.

**Weaknesses**

My major concerns are about the interpretation of the latent distance model, tuning of hyperparameters, and choice of methods for comparison:

1) While the low-rank model (4) is relatively straightforward to interpret, the latent distance model (5) is more difficult to understand. Why does the distance have a negative effect on $\theta_{jk}$? If two features are far apart in the latent space, the relationship implies that they should have a large negative interaction effect. Why is the negative rather than the positive direction preferred? How should one choose the value of $\alpha_0$? The paper should develop some intuition and give examples of applications where this model seems plausible.

2) The proposed optimization problem involves many hyperparameters, including the regularization parameters ($\lambda_1$, $\lambda_2$, and $\lambda_l$) and the latent dimension $d$. Since a fully data-driven method for hyperparameter tuning would be prohibitively expensive, the paper should provide more guidance and heuristics for selecting the hyperparameters.

3) The paper cited only Lou et al. (2013) for sparse modeling of interaction terms. In fact, there are a much larger literature on sparse interaction models in regression. Some early contributions include Zhao et al. (2009), Yuan et al. (2009), Choi et al. (2010), and Radchenko and James (2010). More recent methods allow faster implementations that scale up to 10,000 features (e.g., Wang et al., 2021). These methods should be carefully reviewed and included for numerical comparisons, in addition to the elastic net which does not explicitly model the interaction structures.

**References**

Peng Zhao, Guilherme Rocha, and Bin Yu. The composite absolute penalties family for grouped and hierarchical variable selection. *The Annals of Statistics*, 37(6A):3468-3497, 2009.

Ming Yuan, V. Roshan Joseph, and Hui Zou. Structured variable selection and estimation. *The Annals of Applied Statistics*, 3(4):1738-1757, 2009.

Nam Hee Choi, William Li, and Ji Zhu. Variable selection with the strong heredity constraint and its oracle property. *Journal of the American Statistical Association*, 105(489):354-364, 2010.

Peter Radchenko and Gareth M. James. Variable selection using adaptive nonlinear interaction structures in high dimensions. *Journal of the American Statistical Association*, 105(492):1541-1553, 2010.

Cheng Wang, Binyan Jiang, and Liping Zhu. Penalized interaction estimation for ultrahigh dimensional quadratic regression. *Statistica Sinica*, 31(3):1549-1570, 2021.

---

> ### Author Response · Authors · 2025-08-02
> **Response to Reviewer bGaA**
>
> We thank the reviewer for the feedback and for pointing out relevant related work. Below we address each concern in turn.
>
> # 1. Clarify Interpretation of the Latent Distance Model
> **Comment**: The reviewer is confused about the negative sign in equation 5. “If two features are far apart, why does that imply a large negative interaction effect”
>
> **Response**: The latent distance model was originally introduced by Hoff et al. (2002) for modeling social networks represented by binary adjacency matrices, where two nodes with shorter distances have higher probability of forming an edge (entry = 1), and two nodes with larger distances have lower probability of forming an edge (entry = 0). Accordingly, the distance between nodes has a negative effect on the probability of forming an edge.
>
> This negative effect for distance has also been used for matrices with both positive and negative entries by Huang and Xu (2022) and Nakis et al. (2023). It is also motivated by reasons of interpretability for the HLA compatibility application in this paper, where we want two HLAs that are highly compatible to appear close together in the latent space and two HLAs that are highly incompatible to appear far apart in the latent space. We have added this discussion to the revised paper.
>
> References:
> - Huang, Z., & Xu, K. S. (2022). A latent space model for HLA compatibility networks in kidney transplantation. In Proceedings of the IEEE International Conference on Bioinformatics and Biomedicine (pp. 1020-1027).
> - Nakis, N., Celikkanat, A., Boucherie, L., Djurhuus, C., Burmester, F., Holmelund, D. M., Frolcová, M., & Mørup, M. (2023). Characterizing Polarization in Social Networks using the Signed Relational Latent Distance Model. In 26th International Conference on Artificial Intelligence and Statistics (pp. 11489-11505): Proceedings of Machine Learning Research.
>
> # 2. How to Choose Parameter alpha_0 in equation 5
> **Response**:
> $\alpha_0$ is an intercept term that is estimated. Since distance has a negative effect on the interaction coefficient $\theta_{jk}$, the intercept $\alpha_0$ can be interpreted as the maximum possible positive interaction coefficient, for two features with zero distance in the latent space.
>
> # 3. Give Practical Guidance for Hyperparameter Selection
> **Comment**: Since a fully data-driven method for hyperparameter tuning would be prohibitively expensive, the paper should provide more guidance and heuristics for selecting the hyperparameters.
>
> **Response**:
> Although the model has four hyperparameters, our strategy fixes $d$, uses a limited grid for $\lambda_l$​, and uses typical log scale grids for $\lambda_1, \lambda_2$​. In practice, this keeps the total number of configurations manageable and comparable to standard models like elastic net or factorization machines. See also our response to Reviewer QNQs.
>
> Based on our findings, we suggest:
> - Set $d=2$ as default across different domains
> - Vary $\lambda_l$ over a small grid such as ${0.01, 0.1, 1, 10}$
>
> # 4. Related Work on Sparse Interaction Models
> **Comment**: The authors should review and compare against other methods for sparse modeling of interaction terms.
>
> **Response**: We thank the reviewer for directing us to this related work, which we were not previously aware of. We agree that this related work, particularly methods based on heredity, are relevant because they specifically target interaction terms, particularly by ensuring that the linear terms $x_j$ and $x_k$ are included in the model if the interaction $x_j x_k$ is also included. This represents an improvement over the elastic net with interactions, which does not have this constraint.
>
> Unlike factorization machines and our proposed LIT-LVM approach, these heredity-based methods do not partially tie interaction coefficients together and do not attempt to use other interaction terms $x_{j'} x_{k'}$ to help in estimating $x_j x_k$, which prevents them from “borrowing statistical strength” from other interaction coefficients. Partially tying different interaction coefficients together is particularly important in our kidney transplantation application, as there are many rarely-observed HLA pairs (interaction coefficients with high variance estimates), which are assisted by the estimated interaction coefficients for commonly-observed HLA pairs (low variance estimates).
>
> Due to the short timeframe for the revision, we were not able to add an empirical comparison against a heredity-based method into the paper. However, we have added a discussion of the references you suggested in the related work and limitations sections.
>
> # 5. Minor rephrasing issue
> We have revised the first two sentences (highlighted in blue).

---

> > ### Comment · Reviewer_bGaA · 2025-08-08
> > **Further comments**
> >
> > Thanks for your reply. I have no further comments except for point 1. The latent distance model seems more plausible in logistic and Cox models, where the combined predictors need to pass an exponential transformation, and a larger distance implies a smaller effect after the transformation. But it appears a bit unnatural for linear models, where the combined predictors may change sign as the distance increases. All three references you cited used the model with an exponential link and no features (i.e., $x_{\text{int}}=1$) for the interaction term. The numerical examples in the paper are fine since it is only used with logistic and Cox models.
> >
> > The added discussion is rather difficult to understand: equation (5) specifies how the latent distance affects the interaction effect, not vice versa. For example, it would be better to say “features closer in the latent space should …” rather than “features with [properties of interaction] are placed closer together in the latent space.”

---

### Author Response · Authors · 2025-08-02
**Global Response**

We have revised the paper in accordance with reviewers’ suggestions, with changes highlighted in blue. The main changes are as follows:
- A statement in the second last paragraph of the introduction that clearly indicates where our proposed LIT-LVM approach fits in the space of models.
- A discussion on heredity-based methods for estimating sparse models with interactions in the related work section.
- Additional details motivating the development of the latent distance model in Section 3.1.
- Added discussion of limitations to data with thousands of features in the conclusion, unlike models for ultrahigh-dimensional problems.
- Added Sections C.2 and C.3 in the appendix showing empirical support of convergence of our optimization routine and implementation of sparse factorization machine model using LIT-LVM with extremely high values of $\lambda_l$.

---

### Decision · Action_Editor_Yzrw · 2025-08-21

**Recommendation:** Accept with minor revision

**Additional Comments:**

- As noted by Reviewer `QNQs`, please provide a public reproducible code for the experiments.
- AC Comments: The paper misses some related literature from statistics and corresponding comparisons:
    * How does this relate to hierNet, proposed in *A lasso for hierarchical interactions* from Bien et al. (Ann. Stat., 2013)? Please discuss and also incorporate this in your benchmark comparisons.
    * How does the factorization of additional predictors relate to factorized regression, proposed in *Factorized Structured Regression for Large-Scale Varying Coefficient Models* from Rügamer et al. (ECML, 2022)? Please discuss and also incorporate this in your benchmark comparisons.
    * How does the factorization relate to additive (higher-order) factorization machines, proposed in *Scalable Higher-Order Tensor Product Spline Models* from Rügamer (AIstats, 2024)? Please discuss.

**Audience:**

Yes

**Audience Explanation:**

I think researchers at the intersection of statistics and machine learning (e.g., AIStats) would be interested.

**Claims And Evidence:**

Yes

**Claims Explanation:**

Although some related methods are still missing in the benchmarks (to be addressed in revision), the empirical experiments nevertheless support the claims.